# Cordycepin prevents radiation ulcer by inhibiting cell senescence via NRF2 and AMPK in rodents

Ziwen Wang[1], Zelin Chen[1], Zhongyong Jiang[1], Peng Luo[1], Lang Liu[1,2], Yu Huang[1,2], Huilan Wang[1,6], Yu Wang[1], Lei Long[1], Xu Tan[1], Dengqun Liu[1], Taotao Jin[1], Yawei Wang[1], Yang Wang[1], Fengying Liao[1], Chi Zhang[1], Long Chen[1], Yibo Gan[1], Yunsheng Liu[1], Fan Yang[1], Chunji Huang[3], Hongming Miao[3], Jieping Chen[4], Tianmin Cheng[1], Xiaobing Fu[5] & Chunmeng Shi[1]

The pathological mechanisms of radiation ulcer remain unsolved and there is currently no effective medicine. Here, we demonstrate that persistent DNA damage foci and cell senescence are involved in radiation ulcer development. Further more, we identify cordycepin, a natural nucleoside analogue, as a potent drug to block radiation ulcer (skin, intestine, tongue) in rats/mice by preventing cell senescence through the increase of NRF2 nuclear expression (the assay used is mainly on skin). Finally, cordycepin is also revealed to activate AMPK by binding with the α1 and γ1 subunit near the autoinhibitory domain of AMPK, then promotes p62-dependent autophagic degradation of Keap1, to induce NRF2 dissociate from Keap1 and translocate to the nucleus. Taken together, our findings identify cordycepin prevents radiation ulcer by inhibiting cell senescence via NRF2 and AMPK in rodents, and activation of AMPK or NRF2 may thus represent therapeutic targets for preventing cell senescence and radiation ulcer.

[1] State Key Laboratory of Trauma, Burns and Combined Injury, Institute of Rocket Force Medicine, Third Military Medical University (Army Medical University), 400038 Chongqing, China. [2] Department of Toxicology, Key Laboratory of Environmental Pollution Monitoring and Disease Control, Ministry of Education, Guizhou Medical University, 550025 Guiyang, China. [3] College of Basic Medical Sciences, Third Military Medical University, 400038 Chongqing, China. [4] Department of Hematology, Southwest Hospital, Third Military Medical University, 40038 Chongqing, China. [5] Wound Healing and Cell Biology Laboratory, the First Affiliated Hospital, Chinese PLA General Hospital, Trauma Center of Postgraduate Medical College, 100000 Beijing, China. [6] Institute of Clinical Medicine, Southwest Medical University, 646000 Luzhou, China. Correspondence and requests for materials should be addressed to C.S. (email: shicm@sina.com) or to X.F. (email: fuxiaobing@vip.sina.com)

Radiation ulcer is a common adverse effect of a large dose of radiation for bone marrow transplant or cancer radio-therapy[1–3]. Although radiation therapy technology has progressed substantially, patients still suffer from various degrees of non-specific radiation damage to non-cancerous tissues. The`se chronic wounds can last for several years and cause great distress to patients[4–6]. Up to now, the pathological mechanisms of radiation ulcer remain unsolved and there is no effective medicine[4].

It has been proposed that radiation is a significant contributing factor to DNA damage and cell senescence, which coincide with life-long delayed repair and regeneration of irradiated tissues[7–9]. Cellular senescence is an evolutionarily conserved state of stable replicative arrest induced by pro-ageing stressors also implicated in radiation ulcer pathogenesis, including telomere attrition, oxidative stress, DNA damage and proteome instability[10]. Senescent cells frequently have increased secretion of a broad repertoire of proinflammatory factors, collectively known as the senescence-associated secretory phenotype (SASP), which can induce tissue dysfunction and deterioration in a paracrine manner[11]. However, the relationship between cellular senescence and radiation ulcer remains unclear. Here we hypothesized that senescent cells are involved in radiation ulcer development and preventing cell senescence may represent a prospective strategy to prevent radiation ulcer.

Cordycepin is a natural derivative of adenosine, possessing multiple pharmacological activities including anti-oxidation, anti-tumor, anti-inflammation[12], neuroprotection[13], and preventing bone loss[14]. However, whether cordycepin can prevent radiation-induced cell senescence remains unknown. In this study, we demonstrate that persistent DNA damage foci forms and senescent cells accumulate in radiation ulcer, which are involved in the development of radiation ulcer. Further, we opened an avenue to mitigate radiation ulcer by preventing cell senescence and identified cordycepin as a promising mitigator, acting as a natural NRF2 activator through direct interacting with the α1 and γ1 subunit of AMPK near the autoinhibitory domain which directly relieve autoinhibition.

## Results

### DNA damage and senescent cell accumulate in radiation ulcer.
In this study, in order to identify the relationship between cell senescence and radiation ulcer, a rat skin ulcer model upon high radiation dose (40 Gy) exposure was established[15], which would result in aggravating erythema, desquamation, and ulceration of the skin (Fig. 1a). The fraction of cells with γ-H2AX foci increased sharply within 1 h after rats were irradiated (Fig. 1b, c and Supplementary Fig. 1a–d). Thereafter, the fraction of positive cells gradually declined, but remained significantly elevated over unirradiated controls. Cellular stress induced by persistent DNA damage response is a central mechanism that drives senescence in aging and age-related diseases[16,17]. In irradiated skin sections, we observed sharp increases in SA-β-gal activity 10 days after radiation exposure (Fig. 1d). Similarly, we observed increased levels of the senescence marker p16 and p21 protein after radiation (Fig. 1e). Senescent cells are metabolically active and are able to crosstalk with other cells through several secretory factors, which is also known as SASP[18]. We found that the SASP factors IL1β, IL6 and TNFα significantly increased in irradiated skin compared to non-irradiated tissues (Supplementary Fig. 1e). These results demonstrate that radiation ulcer has long lasting effects in the cells with persistent DNA damage foci, which leads to wide accumulation of senescent cells in tissues.

Further, senescent cells were subcutaneously injected into the irradiated legs. We found senescent cells accelerated the development of radiation ulcer (Fig. 1f). The irradiated legs transplanted with senescent cells became red and swollen at 4 days while the control until the 8th day (Fig. 1f). It began to shed hair on the 4th day in the irradiated legs transplanted with senescent cells which is 4 days earlier than control group (Fig. 1f). And the ulceration of the skin appeared at 8 days in senescent group while the control until the 13th day (Fig. 1f). Therefore, the senescent cells are involved in the development of radiation ulcer and we hypothesized that preventing cell senescence could be a therapeutic strategy to mitigate radiation ulcer.

### Cordycepin decreases cell senescence and SASP in vitro.
Based on above, we first established an in vitro fibroblast cell senescence model induced by radiation to screen the candidate agents with a special attention on chemical small molecules which can inhibit cell senescence since fibroblasts play vital roles in the development of ulcer. Next, we used a library of small molecules established in our group to screen the potential radio-protective candidates for further investigations. Among the small molecules tested, we identified cordycepin, a natural nucleoside analogue compound, as an effective mitigator to reduce the cell senescence and radiation-induced ulcer (Fig. 2d, e and Supplementary Fig. 2a–e). To further identify whether cordycepin could rescue the deleterious effects of ionizing radiation in vitro, we treated skin fibroblasts with 200 μM cordycepin for 3 days before irradiation and found cordycepin-treated fibroblasts exposed to radiation showed reduced levels of the DNA damage response marker γ-H2AX at early and late time points (Fig. 2a–c). The activation of the DNA damage response is one of the main mediators of cell senescence, which results in irreversible growth arrest[19,20]. Accordingly, we observed reduced levels of the senescence marker of p16 and p21 proteins in cordycepin-treated fibroblasts (Fig. 2d) and human keratinocytes (HaCaT cells) after radiation (Supplementary Fig. 2a). Similarly, SASP and SA-β-gal activity were both decreased in fibroblasts (Fig. 2e and Supplementary Fig. 2b, c) and HaCaT cells (Supplementary Fig. 2d, e). These findings suggested that cordycepin can prevent cells from entering senescence by reducing DNA damage and the consequent induction of senescence and SASP, therefore increasing replicative capacity of cells. Following radiation, the proliferation of cells is largely decreased, however, we found cordycepin improved the proliferation after radiation (Supplementary Fig. 2f). In addition, cordycepin prevents radiation-induced apoptosis (Fig. 2f). We also used $H_2O_2$ to mimic cell damage caused by oxidative stress in vitro, fibroblasts preconditioned with cordycepin demonstrated better resistance to acute oxidative stress with less apoptosis (Fig. 2g). In conclusion, cordycepin improves the cell proliferation and reduces cell death after oxidative stress.

Since each large colony is expected to grow from a single surviving self-renewing stem cell[21]. We found cordycepin increased the colony forming efficiency of irradiated fibroblasts and HaCaT cells (Fig. 2h and Supplementary Fig. 2g). These findings indicate that cordycepin increases the survival and repopulating capacity of cells, hence protecting from radiation-induced loss of regenerative cell population. On the other hand, reactive oxygen species (ROS) can mediate the deleterious effects of radiation and contribute to DNA damage and the activation of senescence pathways in cells[15,19], and we found that treatment with ROS scavenger N-acetylcysteine (NAC) at least partially abolished the radiation-induced cell senescence (Supplementary Fig. 2b, c). By analyzing the levels of cytosolic and mitochondrial ROS, we found that cordycepin treatment prevented the induction of ROS after radiation in fibroblasts (Fig. 2i). Cytosolic

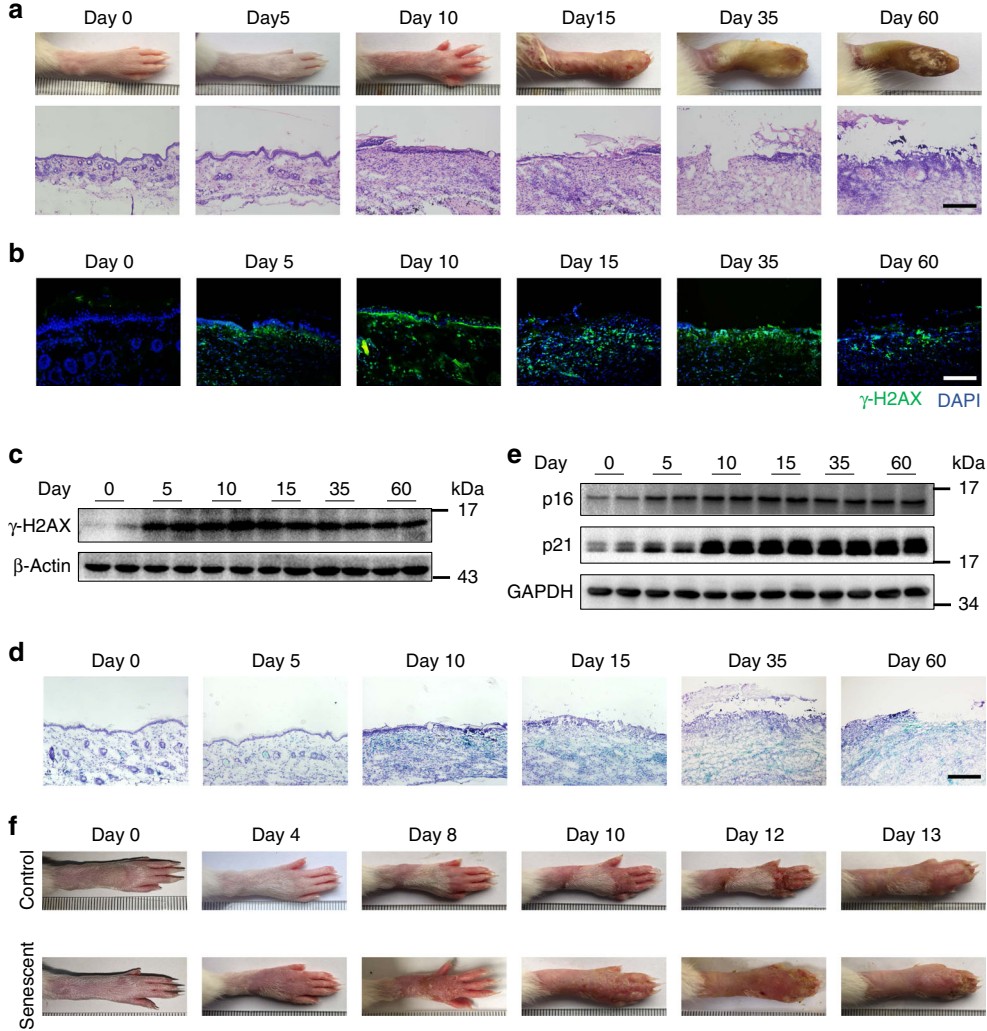

**Fig. 1** DNA damage and senescent cell accumulate in radiation ulcer. **a** Representative images of hind limb 0–60 days post-radiation (top), representative histological analysis of rat skin tissues 0–60 days post-radiation (bottom). **b** Representative immunofluorescence pictures of γ-H2AX in rat skin tissues 0–60 days post-radiation. **c** Western blot analysis of γ-H2AX expression in rat skin tissues 0–60 days post-radiation. **d** Representative SA-β-gal staining in rat skin tissues 0–60 days post-radiation. **e** Western blot analysis of p16 and p21 levels in rat skin tissues 0–60 days post-radiation. **f** Representative images of radiated-hind limb after subcutaneously injected the senescent cells with an equal volume of growth factor-reduced Matrigel or control. This test was repeated three times. Representative images were shown. Bars represent 200 μm (bottom) (**a**), 100 μm (**b**), 200 μm (**d**)

ROS of HaCaT cells was also reduced in cordycepin-treated group (Supplementary Fig. 2h). Then we measured the levels of different antioxidants involved in the scavenging of ROS and found the expression of SOD1, SOD2, GPX-1, and Catalase in fibroblasts were all increased at different time after cordycepin treatment (Fig. 2j). Together, these findings suggest that cordycepin may increase the clonal proliferative capacity and antioxidant ability of cells and prevent the activation of specific cell senescence mechanisms.

**Cordycepin prevents radiation ulcer.** Considering the remarkable effects of cordycepin in vitro, we next tested if cordycepin could prevent radiation ulcer. Three kinds of ulcer models were established to evaluate the effect of drugs, including rat skin, mouse intestine and mouse oral ulcer. Interestingly, we identified that cordycepin efficiently prevented radiation-induced cutaneous ulcer and oedema, and increased dermal cell restore and re-epithelialization (Fig. 3a, b). One of the long-term effects post-radiation is fibrosis and microscopic analysis of the skin from cordycepin-treated group did not reveal significant fibrosis with trichrome staining compared with control group (Supplementary

Fig. 3a). In radiation-induced intestine ulcer, we found that cordycepin-treated group displayed better-preserved intestinal structure (Fig. 3c), a two-fold-increase in surviving crypts and taller villi (Fig. 3d). All vehicle-treated mice died 5–7 days after radiation, in contrast, 50–60% of mice in the cordycepin-treated group (the survive rate in prevention and treatment group are 57 and 48%, respectively) survived at least 40 days after radiation (Fig. 3e) and we didn't find the development of fibrosis in the intestine tissue in mice survived (Supplementary Fig. 3b). Meanwhile, cordycepin almost completely prevented the appearance of mucositis/ulcers in irradiated tongue (Fig. 3f and Supplementary Fig. 3c). Histological analysis of the tongues revealed the preservation of the epithelial layer in irradiated cordycepin-treated mice, although radiation-associated tissue changes were observed (Fig. 3g). Further, H&E staining showed that the lung, liver, spleen, heart, kidney and intestine were not significantly influenced by cordycepin (Supplementary Fig. 12). Inhibiting the mammalian target of rapamycin (mTOR) with rapamycin has been reported to increase the clonogenic capacity of primary human oral keratinocytes and their resident self-renewing cells by preventing stem cell senescence, mTOR

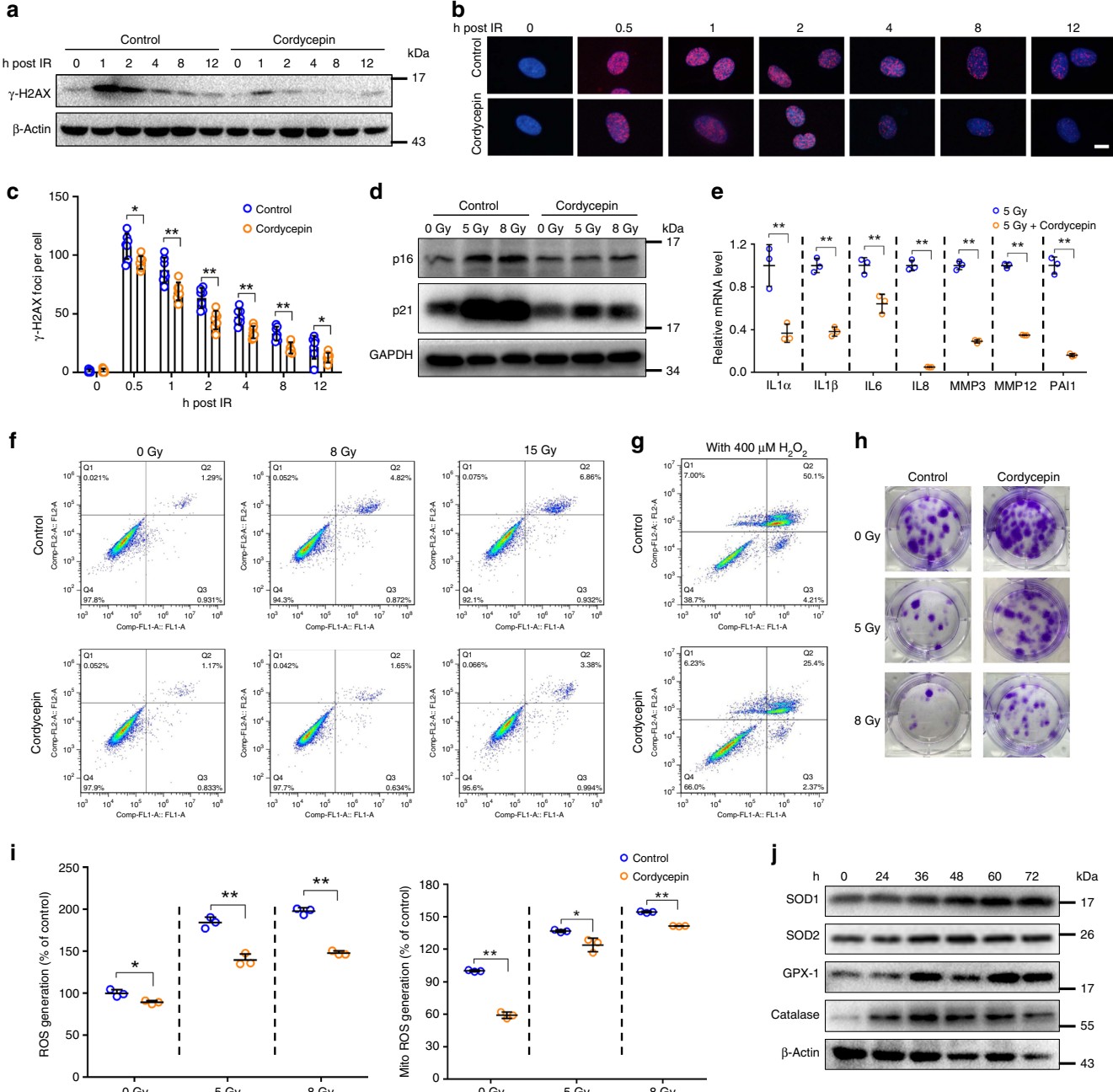

**Fig. 2** Cordycepin decreases cell senescence and SASP in vitro. **a** Western blot analysis of γ-H2AX expression in fibroblasts after 5 Gy radiation. **b** Representative images of nuclei γ-H2AX in fibroblasts 0–12 h after 5 Gy radiation. This test was repeated three times. Representative images were shown. **c** Average number of γ-H2AX per nucleus 0–12 h after 5 Gy radiation ($n = 7$). **d** Western blot analysis of p16 and p21 levels in irradiated control or cordycepin-treated fibroblasts 7 days after radiation. This test was repeated three times. Representative images were shown. **e** Quantification of mRNA expression for senescence secretory phenotype in irradiated control or cordycepin-treated fibroblasts 7 days after radiation ($n = 3$). **f** Fibroblasts were treated with cordycepin for 72 h and harvested for detection of apoptosis using flow cytometry 24 h following exposure to radiation. **g** Fibroblasts were treated with cordycepin for 72 h and harvested for detection of apoptosis using flow cytometry 12 h following $H_2O_2$ treatment. **h** Representative images of fibroblasts colonies generated in survival assays following 0/5/8 Gy radiation. **i** Analysis of reactive oxygen species (ROS) levels and mitochondrial superoxide by H2DCF-DA or MitoSOX Red staining 24 h after radiation in fibroblasts pretreated or not (control) with cordycepin ($n = 3$). **j** SOD1, SOD2, GPX-1, and Catalase expression levels in whole cell lysates from cordycepin-treated fibroblasts for the indicated times. Bars represent 10 μm (**b**). Data in **c**, **e**, and **i** represent the means ± S.D. (*$P < 0.05$, **$P < 0.01$; student's $t$-test)

inhibition also protects from the loss of proliferative basal epithelial stem cells upon ionizing radiation in vivo, thereby preserving the integrity of the oral mucosa and protecting from radiation-induced mucositis[22]. However, rapamycin did not protect from mucositis compared with cordycepin when mice were irradiated in a single dose (15 Gy) (Supplementary Fig. 3d),

indicating that the protection conferred by cordycepin treatment might be more effective than rapamycin in a high single dose.

**Cordycepin prevents the loss of cell proliferation in vivo.** Further study showed a remarkable reduction in the levels of γ-H2AX in irradiated skin, intestine and tongue (Fig. 4a–c),

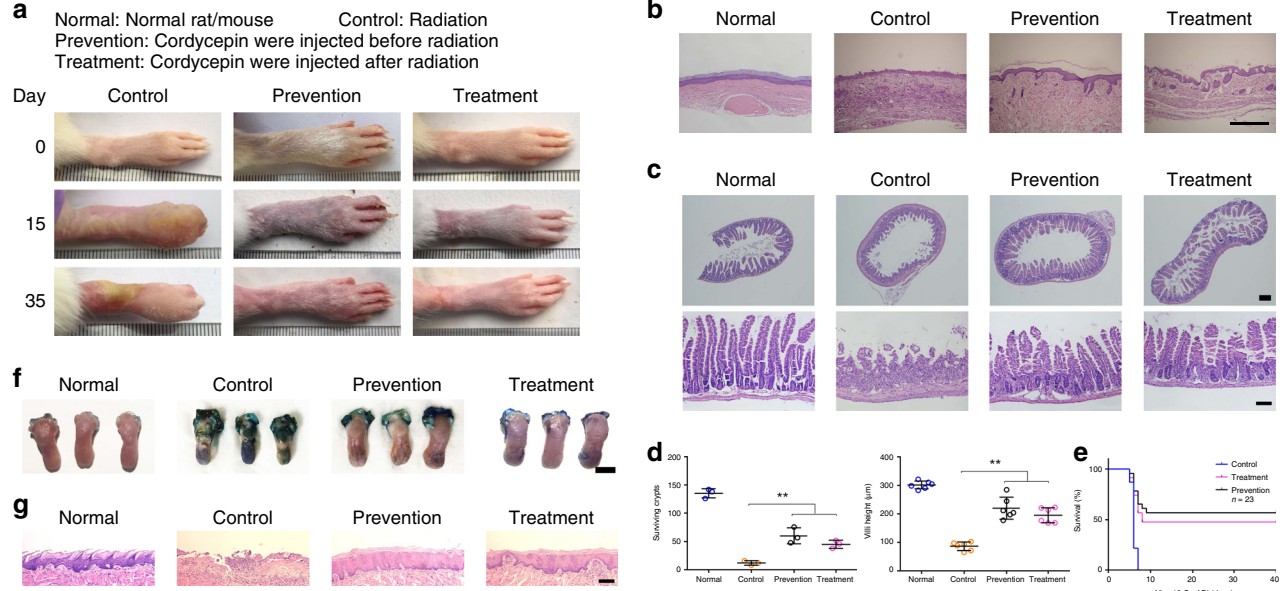

**Fig. 3** Cordycepin prevents radiation ulcer. **a** Images of hind limb from irradiated rats (control, prevention, treatment) on day 0, 15, and 35. This test was repeated three times. Representative images were shown. Control: Rat right posterior limb (other parts covered with lead board) was exposed to 40 Gy radiation under anesthesia; Prevention: Cordycepin were injected before radiation; Treatment: Cordycepin were injected after radiation. **b** Histological analysis of the skins from normal rats and irradiated rats (control, prevention, treatment) 35 days after radiation. **c** Histological analysis of intestinal sections from normal mice and irradiated mice (control, prevention, treatment) 4 days after radiation. This test was repeated three times. Representative images were shown. Control: Mouse abdominal (other parts covered with lead board) was exposed to 12 Gy radiation under anesthesia. Prevention: Cordycepin were injected before radiation; Treatment: Cordycepin were injected after radiation. **d** Quantitation of surviving crypts ($n = 3$) and villus height ($n = 6$) from normal mice and irradiated mice (control, prevention, treatment) 4 days after radiation, the villus height was calculated using Image J software. **e** Survival curve of mice subjected to abdominal radiation ($n = 23$). **f** Representative pictures of tongues stained with toluidine blue from normal mice and irradiated mice (control, prevention, treatment) 10 days after radiation. Lack of protective epithelial barrier and therefore ulcer formation is indicated by deep, royal blue staining in epithelium defects. This test was repeated three times. Representative images were shown. Control: Mouse head and neck area (other parts covered with lead board) was exposed to 15 Gy radiation under anesthesia. Prevention: Cordycepin were injected before radiation; Treatment: Cordycepin were injected after radiation. **g** Histological analysis of the tongues from normal mice and irradiated mice (control, prevention, treatment) 10 days after radiation. Bars represent 500 μm (**b**), 500 μm(top) and 100 μm(bottom) (**c**), 300 μm (**f**), 100 μm (**g**). Data in **d** and **e** represent the means ± S.D. (**$P < 0.01$; student's $t$-test)

indicating that cordycepin may protect cells from DNA damage and oxidative stress in vivo. Aligned with our in vitro results, we observed substantial differences between control and cordycepin-treated mice/rats in irradiated skin, intestine and tongue by TUNEL assays (Supplementary Fig. 4a–c). On the other hand, we found cordycepin significantly decreased radiation-induced SA-β-gal⁺ cells, levels of the senescence marker p16, p21 protein, and the secretion of senescence-associated cytokines in irradiated skin (Fig. 4d and Supplementary Fig. 5a, b). Similarly, qRT-PCR showed that cordycepin significantly decreased radiation induced senescence marker p16, p21, and the SASP of intestine and tongue (Supplementary Fig. 5c–f). Moreover, cordycepin dramatically reduced radiation-induced SA-β-gal⁺ cells in both intestine and tongue models (Supplementary Fig. 6a, b). Plasma concentrations of IL-1β, IL-6, and TNF-α were significantly lower in cordycepin-treated rats/mice than controls due to high dose radiation in both skin and intestine, which often causes systemic effects (Supplementary Fig. 6c, d). By using the proliferation marker Ki-67 or BrdU, we found that cordycepin prevented the radiation-induced block in proliferation of the basal layer of epidermis, crypts and basal progenitor layer of the oral epithelia (Fig. 5a–c). As shown in Fig. 5d, we found that cordycepin prevented the radiation-induced disappearance of the proliferation and epithelial progenitor marker p63 in oral epithelia, indicating that cordycepin reduces the depletion of stem cells from radiation. These results highlight the possibility that cordycepin may increase the proliferative capacity by preventing the senescence of their repopulating stem cells in vivo.

**Cordycepin prevents senescence which is NRF2-dependent.** In search for the potential mechanisms mediating the protective effect of cordycepin in cell senescence, we explored whether cordycepin treatment of fibroblasts results in the activation of molecular events was often associated with delayed aging. After cordycepin treatment, however, we did not observe an increase in the expression, nuclear localization, or transcriptional activation of FoxO proteins, and changes in the expression of SIRT family members (Supplementary Fig. 7a–c), all of which have been shown to influence lifespan and stem cell response to stress[23,24]. Nuclear factor erythroid 2-related factor 2 (NRF2) is a critical redox sensor and is one of the master regulators of antioxidant responses. NRF2 binds to the antioxidant response elements (AREs) and activates the transcription of a number of antioxidant genes that is known for counteracting ROS[25]. And we found a significant increase in the expression of NRF2 in fibroblasts (Fig. 6a) and HaCaT cells (Supplementary Fig. 7d). The nuclear localization (the next step in NRF2 pathway activation) of NRF2 was significantly increased in cordycepin-treated fibroblasts while the cytosolic NRF2 fraction remained unchanged (Fig. 6b and Supplementary Fig. 7e). At the same time, immunofluorescence staining of fibroblasts showed that cordycepin increased the amount of NRF2 in the nucleus (Fig. 6c). Moreover, cordycepin

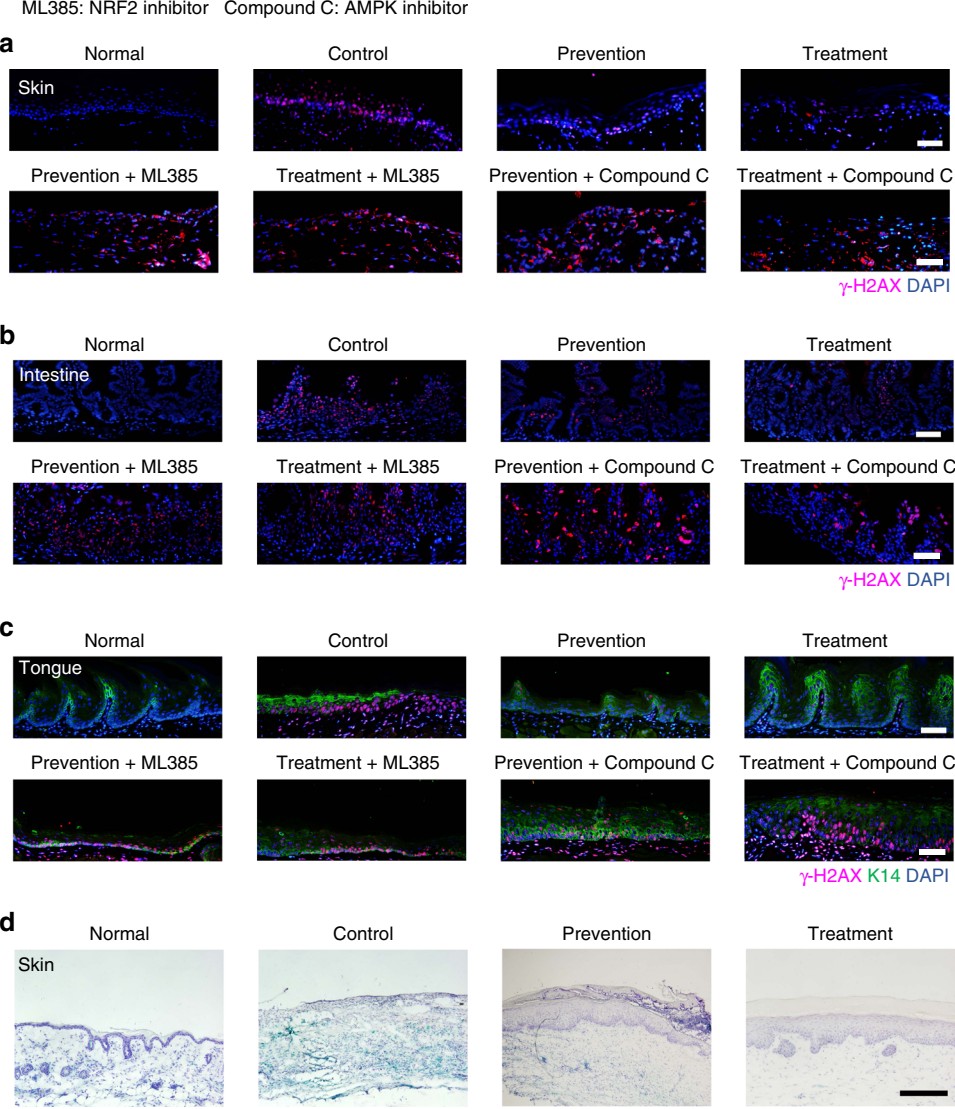

**Fig. 4** Cordycepin prevents DNA damage and cell senescence in vivo. **a–c** Representative immunofluorescence pictures and quantification of γ-H2AX in rat skin, mouse intestine and mouse tongue sections from normal animals and irradiated animals 35 days after irradiation, respectively. Basal progenitor marker cytokeratin 14 (K14) was used to label the basal progenitor layer of tongue. The γ-H2AX positive nuclei was calculated using Image J software. **d** Representative SA-β-gal staining in skin tissues from normal rats and irradiated rats 35 days after radiation. ML385: NRF2 inhibitor; Compound C: AMPK inhibitor. Bars represent 50 μm (**a–c**), 200 μm (**d**)

resulted in an increase in nuclear NRF2 activity in cordycepin-treated animals (Fig. 6d). Phosphorylation on serine 40 is a regulatory modification implicated for NRF2 translocation to the nucleus and downstream protein activation[15,26,27]. The phosphorylation of NRF2 on serine 40 in the nuclear compartment of fibroblasts was increased by cordycepin treatment (Fig. 6e). No change was observed in phosphorylation of cytosolic NRF2 in fibroblasts (Supplementary Fig. 7f). We used lamin and actin respectively as markers for purified nuclear and cytosolic fractions.

We next quantified the relative amounts of thioredoxin (TXN), which reduces the cysteine residues critical for binding of nuclear NRF2 to ARE[28], and found a significant TXN increase in cordycepin-treated fibroblasts (Supplementary Fig. 7g). To determine the functional relevance of the increases in expression, translocation and phosphorylation of NRF2, we measured NRF2 transcriptional activity using a luciferase-based ARE-controlled gene expression system. Cordycepin increased luciferase activity significantly compared to untreated cells (1–3 folds)

(Supplementary Fig. 7h). To further confirm NRF2 activation, we next measured the relative expression of NRF2 downstream target genes, Srx, NAD(P)H dehydrogenase quinone 1 (NQO1), and glutathione S-transferase A1 (GSTA1) by qRT-PCR in fibroblasts treated with cordycepin at different time and they were all upregulated (Fig. 6f). At the same time, consistent with in vitro results, the NRF2 target genes from the cordycepin-treated mouse/rat models are all increased (Supplementary Fig. 7i). We also measured the relative abundance of NRF2 downstream targets in fibroblasts, GCLC, GCLM, HMOX1, and NQO1 by western blot analysis, all four targets showed a significant increase in expression (Fig. 6g).

To determine the role of NRF2 in the cordycepin-induced inhibition of cell senescence, we investigate the effect of cordycepin on fibroblasts by knockdown of NRF2. However, cells deficient in NRF2 have more p16, p21 protein levels, higher level of senescent cells and SASP although treated with cordycepin (Fig. 6h–j and Supplementary Fig. 7j). NRF2 knockdown reverted the protective effects of cordycepin in fibroblast

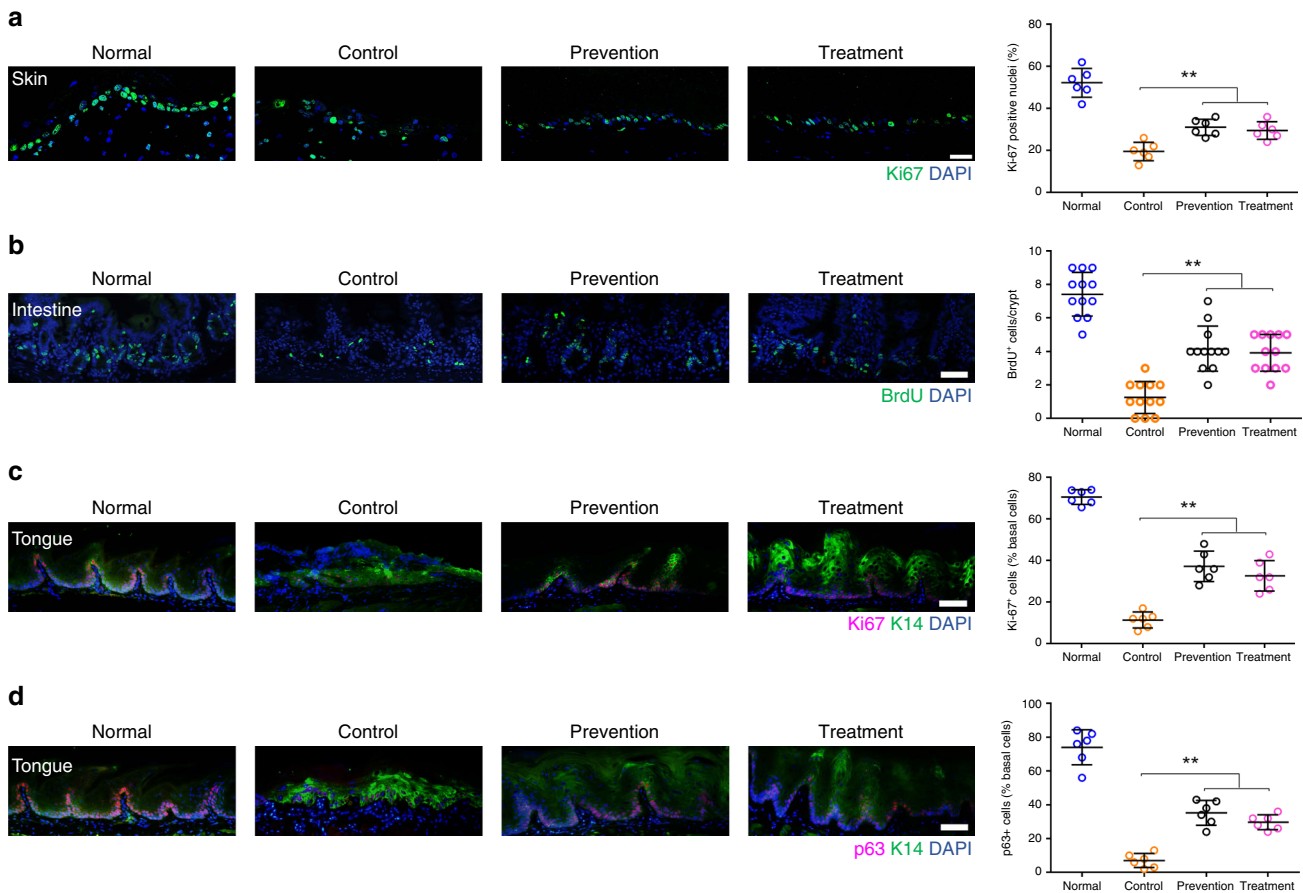

**Fig. 5** Cordycepin prevents the loss of proliferation in vivo. **a** Representative confocal pictures and quantitation of Ki-67 in skin sections from normal rats and irradiated rats (control, prevention, treatment) 35 days after irradiation ($n = 6$). **b** Representative immunofluorescence pictures and quantitation of BrdU in intestine sections from normal mice and irradiated mice (control, prevention, treatment) 4 days after irradiation. The BrdU positive nuclei was calculated using Image J software ($n = 12$). **c, d** Representative immunofluorescence pictures and quantitation of Ki67$^+$ and p63$^+$ cells (% basal cells) in tongue sections from normal mice and irradiated mice (control, prevention, treatment) 10 days after irradiation, basal progenitor marker cytokeratin 14 (K14) was used to label the basal progenitor layer of tongue ($n = 6$). The Ki-67 and p63 positive nuclei was calculated using Image J software. Bars represent 25 μm (**a**), 50 μm (**b–d**). Data in **a–d** represent the means ± S.D. (**P < 0.01; student's *t*-test)

clonogenic capacity (Fig. 6k) and reverted the decrease in cytosolic ROS generation (Fig. 6l). These data suggest that the cordycepin-induced inhibition of cell senescence and the inflammatory phenotype are both NRF2 dependent.

To further identify the underlying mechanisms of cordycepin in vivo, we applied a newly developed strategy using specific NRF2 inhibitor (ML385) to inhibit the activation of NRF2 in our study[29,30]. Using the western blotting, we found that ML385 effectively inhibited the NRF2 activity in vivo (Supplementary Fig. 8a), and H&E staining showed that the lung, liver, spleen, heart, kidney and intestine were not significantly influenced by ML385 (Supplementary Fig. 12). Interestingly, ML385 significantly blocked the protective effects of cordycepin in radiation induced skin, intestine and tongue ulcers (Supplementary Fig. 8c, d and Supplementary Fig. 3d). In radiation-induced intestine ulcer model, the survival rates in cordycepin with ML385 group were similar to those of the control group (Supplementary Fig. 3e). Since radiation ulcer has long lasting effects in the cells with persistent DNA damage foci, which leads to a tissue wide accumulation of senescent cells, we found NRF2 inhibition reversed the γ-H2AX levels in irradiated skin, intestine and tongue (Fig. 4a–c). Similarly, the levels of the senescence marker and SASP phenotype were increased significantly when treated with ML385 (Supplementary Fig. 5a–f). Therefore, we further

verified that cordycepin mitigates the radiation ulcer is NRF2-dependent.

**Cordycepin promotes autophagic degradation of Keap1.** We next asked how cordycepin regulates NRF2 activity. Cordycepin led to significant induction of NRF2 expression at the protein, but not the mRNA level (Fig. 6a and Fig. 7a). These data suggest that cordycepin induces NRF2 accumulation, and that this is unlikely to occur at the transcriptional level. Therefore, we postulated that cordycepin might elevate NRF2 by regulating its protein stability. To test this hypothesis, we treated cells with the protein synthesis inhibitor cycloheximide (CHX). The NRF2 protein level gradually decreased in cordycepin-treated fibroblasts after CHX treatment in a time-dependent manner as expected, however, more striking degradation of NRF2 was observed in control cells (Fig. 7b). By analyzing the quantification curve, we found that the half-life of NRF2 appeared to be around 20 min (min) in control cells and 80 min in cordycepin-treated cells (Fig. 7b), suggesting that cordycepin is critical to maintain NRF2 stability.

Under basal conditions, NRF2 activity is tightly restricted by binding with Kelch-like ECH associated protein 1 (Keap1) in the cytoplasm. We found cells preconditioned with cordycepin showed reduced abundance of Keap1 in a time-dependent manner in fibroblasts and HaCaT cells (Fig. 7c and

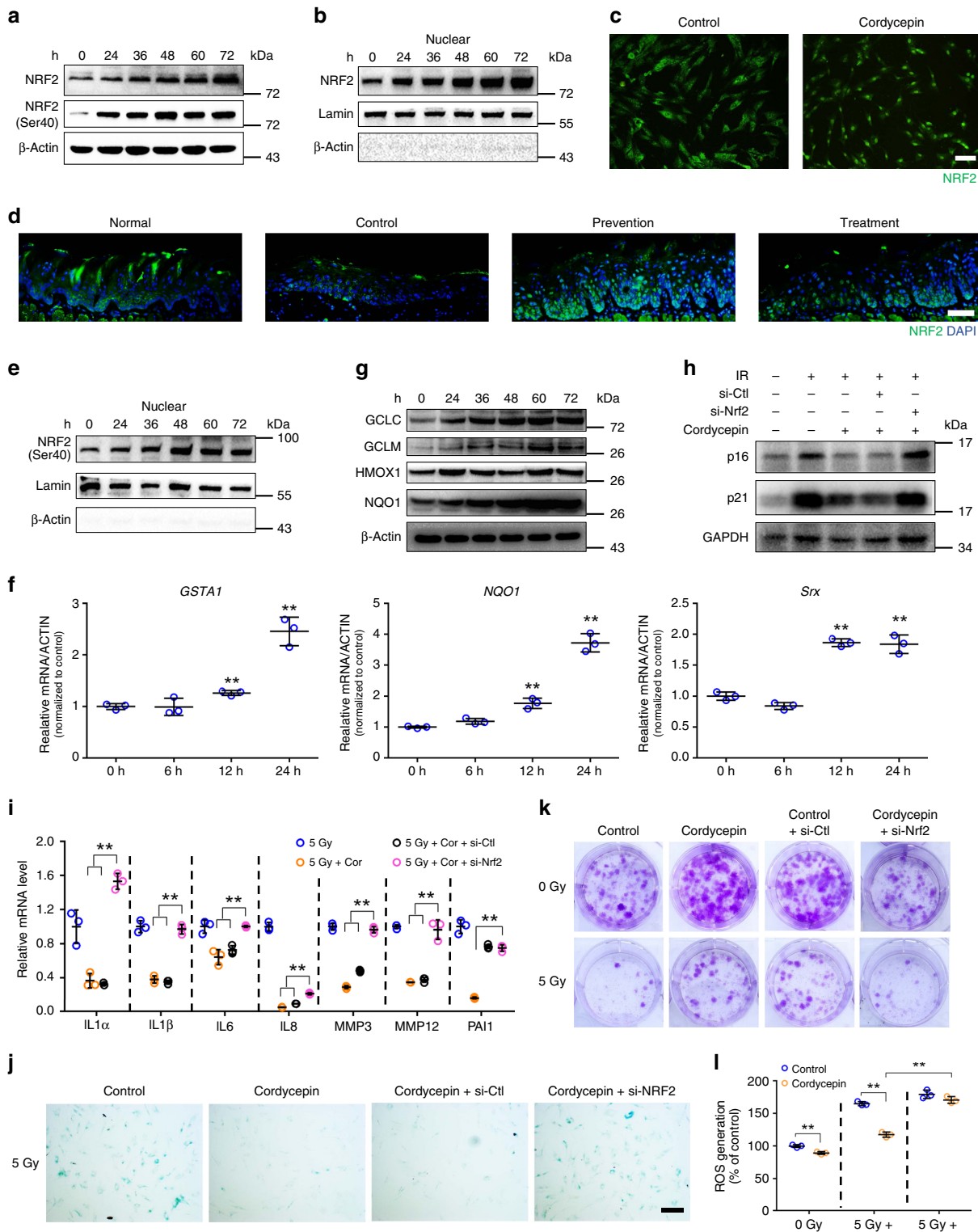

Supplementary Fig. 9a), whereas the amount of Keap1 mRNA remained unaffected (Supplementary Fig. 9b). Moreover, cordycepin also resulted in a decrease in Keap1 activity in cordycepin-treated animals (Fig. 7d). Therefore, we hypothesize that the downregulation of Keap1 abundance by cordycepin is mediated by degradation of Keap1. Two major pathways mediate protein degradation: the ubiquitin-proteasome and autophagy-lysosome pathways. Inhibition of the proteasome by MG-132, a proteasome inhibitor, had no effect on keap1 abundance (Fig. 7e), whereas

cordycepin-induced decrease in the abundance of Keap1 was attenuated by distinct autophagy inhibitors : chloroquine (CQ) (Fig. 7f and Supplementary Fig. 9c) and 3-methyladenine (3MA) (Fig. 7g and Supplementary Fig. 9d), suggesting that the downregulation of Keap1 by cordycepin is likely mediated via autophagic degradation. To determine whether cordycepin stimulates autophagy, we examined the lipidation of LC3, a characteristic of autophagy that converts the LC3-I isoform to the electrophoretically more mobile LC3-II isoform and found

**Fig. 6** Cordycepin prevents senescence and SASP, which are NRF2-dependent. **a** NRF2 and p-NRF2 expression levels in whole cell lysates from cordycepin-treated fibroblasts for the indicated times. This test was repeated three times. Representative images were shown. **b** NRF2 expression levels in nuclei from cordycepin-treated fibroblasts for the indicated times. **c** Representative immunofluorescence pictures of NRF2 in control and cordycepin-treated (3 days) fibroblasts. This test was repeated three times. Representative images were shown. **d** Representative immunofluorescence pictures of NRF2 of the tongues from normal mice and irradiated mice (control, prevention, treatment) 10 days after radiation. **e** p-NRF2 expression levels in nuclei from cordycepin-treated fibroblasts for the indicated times. **f** Quantification of mRNA expression for GSTA1, NQO1 and Srx from cordycepin-treated fibroblasts for the indicated times. **g** GCLC, GCLM, HMOX1 and NQO1 expression levels in whole cell lysates from cordycepin-treated fibroblasts for the indicated times. **h** Western blot analysis of p16 and p21 levels in irradiated control or cordycepin-treated (3 days) fibroblasts following knockdown of NRF2 7 days after radiation. **i** Quantification of mRNA expression for senescence secretory phenotype in fibroblasts following knockdown of NRF2 7 days after radiation. **j** Staining for SA-β-gal 7 days after radiation in fibroblasts following knockdown of NRF2. This test was repeated three times. Representative images were shown. **k** Representative images of fibroblasts colonies generated in survival assays following knockdown of NRF2. **l** Analysis of reactive oxygen species (ROS) levels 24 h after radiation in fibroblasts following knockdown of NRF2. Bars represent 50 μm (**c**), 100 μm (**d**), 250 μm (**j**). Data in **f**, **i**, and **l** represent the means ± S.D. (n = 3, *P < 0.05, **P < 0.01; student's t-test)

cordycepin-induced expression of the LC3-II autophagy marker in fibroblasts (Fig. 7h and Supplementary Fig. 9e) and HaCaT cells (Supplementary Fig. 9a). TEM images showed autophagic vacuole formation in cordycepin-treated fibroblasts (Fig. 7i). Consistent with this interpretation, the role of autophagy was further studied. Depletion of the autophagy components ATG7 also reverted the decrease of keap1 protein (Fig. 7j).

Autophagy can selectively degrade certain substrates, mediated by specific autophagic adaptors[31,32]. Several pathways that regulate the Keap1-NRF2 interaction have been identified, one of which involves p62, an autophagy substrate that competes with NRF2 for binding to Keap1[33,34]. Indeed, depletion of the autophagic adaptor p62 blocked Keap1 degradation induced by cordycepin (Fig. 7k), suggesting that p62-dependent autophagy is the major pathway for such degradation. On the other hand, p62 knockdown reverted the decrease in p16, p21 protein, and SA-β-gal+ fibroblasts (Supplementary Fig. 9f–h). Thus, keap1 appears to be a target for autophagic degradation through its association with p62 when treated with cordycepin, then promoting the accumulation of NRF2 protein.

To exclude the possibility that reduction of Keap1 protein level is marginal and to further demonstrate that cordycepin activates NRF2 is attributed to the Keap1 degradation, fibroblasts were transfected with the KEAP1 expressing plasmid and we found that the NRF2 protein level was reversed after treated with cordycepin (Supplementary Fig. 9i), suggesting that cordycepin activates NRF2 in a keap1-depedent manner.

**Cordycepin interacts with α1 and γ1 subunit of AMPK.** To explore the potential mechanisms by which cordycepin activates autophagy, we examined the signaling pathways regulated by cordycepin treatment. PI3K/Akt/mTOR pathway is among the most common pathways in the process of autophagy[35]. Treatment with cordycepin decreased mTOR activity as assessed by the phosphorylation of S6 in fibroblasts and HaCaT cells (Fig. 8a and Supplementary Fig. 10a). Moreover, cordycepin also resulted in a decrease in S6 activity in cordycepin-treated animals (Fig. 8b and Supplementary Fig. 10b). But we failed to detect any obvious change in AKT activity in fibroblasts (Fig. 8a), suggesting that the PI3K/AKT pathway might not be involved in the autophagy caused by cordycepin. According to former studies[36], phosphorylation of AMPK causes inactivation of mTOR and then induces autophagy. As shown in Fig. 8c, treatment with cordycepin significantly increased the levels of phospho-AMPK and its downstream substrate, acetyl-CoA carboxylase (ACC) in fibroblasts. We also found elevated AMPK expression in cordycepin-treated HaCaT cells (Supplementary Fig. 10a). To verify the role of AMPK in cordycepin-caused autophagy, we pretreated fibroblasts with Compound C, an AMPK inhibitor, results showed that combined treatment of Compound C downregulated AMPK

and upregulated mTOR activity as well as reduced trans-version of LC3B-I to LC3B-II (Fig. 8d). Therefore, the AMPK activation is responsible for the cordycepin-induced autophagy process. There is a report that AMPK phosphorylates NRF2 at the Ser550 residue, which in conjunction with AMPK-mediated GSK3-β inhibition, promotes nuclear accumulation of NRF2 for anti-oxidant response element (ARE)-driven gene transactivation[37]. However, we didn't find significant increase of GSK3-βSer9 (Supplementary Fig. 10c), we therefore believe that AMPK-dependent degradation of Keap1 is the major pathway for cordycepin to activate NRF2.

We next verified the role of AMPK in cordycepin-induced cytoprotective ability, supplementation with Compound C reverted the decrease in ROS generating (Fig. 8e) and the protective effects of cordycepin in fibroblasts clonogenic capacity under basal conditions and after radiation exposure (Fig. 8f). On the other hand, AMPK inhibition also reverted the decrease in p16, p21 protein and SA-β-gal+ fibroblasts (Fig. 8g and Supplementary Fig. 10d). Furthermore, we used Compound C to inhibit AMPK in vivo (Supplementary Fig. 8b), H&E staining showed that the lung, liver, spleen, heart, kidney and intestine were not significantly influenced by Compound C (Supplementary Fig. 12), the protective effects of cordycepin in radiation induced skin, intestine and tongue ulcers were significantly reversed (Supplementary Fig. 8c, d and Supplementary Fig. 3d), the survival rates in cordycepin with Compound C group were like those of the control group in radiation-induced intestine ulcer model (Supplementary Fig. 3e). On the other hand, the γ-H2AX levels in irradiated skin, intestine and tongue were reversed when AMPK was inhibited by compound C (Fig. 4a–c).

Similarly, the levels of the senescence marker and SASP phenotype were increased significantly when treated with ML385 (Supplementary Fig. 5a–f).

Increasing the intracellular AMP/ATP ratio and stimulation of upstream kinases such as LKB1 and CaMKK2 are the two primary activators of AMPK[38,39]. As shown in Fig. 8h, neither intracellular concentrations of AMP, ADP and ATP nor the AMP/ATP ratio was significantly changed after treatment with cordycepin. On the other hand, we did not find obvious changes in activation of LKB1 and CAMKK2 (Supplementary Fig. 10c). Supplementation with STO-609, a selective CaMKK2 inhibitor, did not inhibit cordycepin-induced AMPK activation either (Supplementary Fig. 10e), suggesting that cordycepin can activate AMPK in the absence of CaMKK2. When the expression of LKB1 was decreased by specific siRNAs, the stimulatory effect of cordycepin on AMPK was as before (Supplementary Fig. 10f). Taken together, these results indicate that cordycepin can promote AMPK activity directly and not as a consequence of stimulation of upstream kinases, or by altering the AMP/ATP ratio. Therefore, we hypothesized that

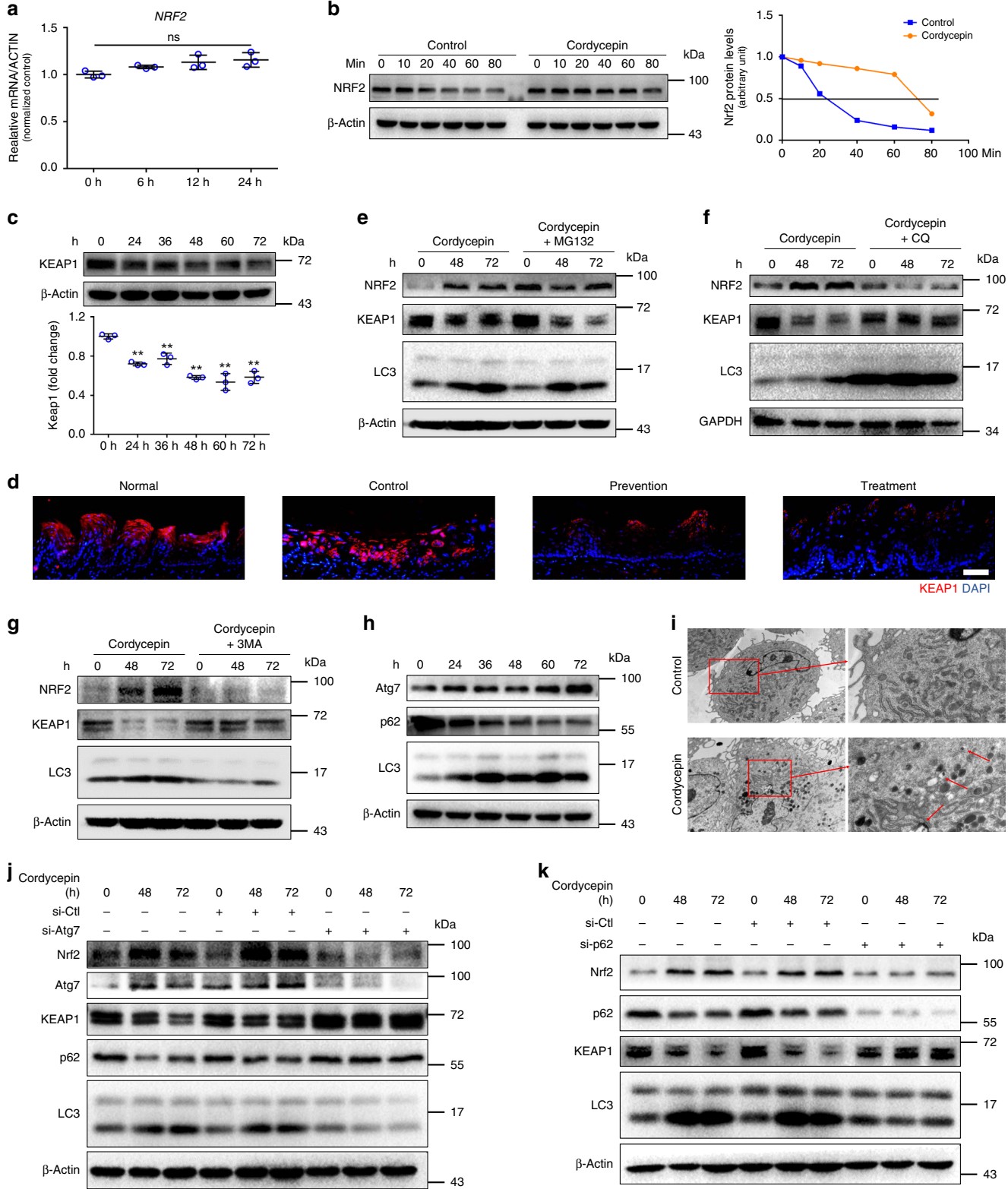

cordycepin can activate AMPK by directly binding to AMPK protein.

Next, we investigated the interaction between cordycepin and AMPK. AMPK exists as a heterotrimeric complex comprised of a catalytic α (α 1/2) subunit and non-catalytic, regulatory β (β 1/2) and γ (γ 1/2/3) subunits that differ in their subcellular localization and AMP-dependence. The binding ability of cordycepin to each subunit of AMPK was evaluated by molecular docking using

POCASA. Cordycepin cannot bind steadily to the α2, γ2, γ3, β1 or β2 subunits of AMPK, but binds to α1 and γ1 subunit near the autoinhibitory domain (AID) with relative high affinity. As displayed in Fig. 8i, there are two binding sites near AID, site 1 locates in the contact area between α1 and γ1 domain, site 2 is completely located in α1 domain. Moreover, the residues Asp331, Asn332, Phe32, and Ser35 are key residues corresponding for the binding of site1, the residues Asp292, Asp282, Leu313, and

**Fig. 7** Cordycepin activates NRF2 by promoting autophagic degradation of Keap1. **a** Quantification of mRNA expression for NRF2 from cordycepin-treated fibroblasts for the indicated times. **b** Western blot analysis of NRF2 levels in the control and cordycepin-treated cells in the presence of 100 mg/mL CHX for the indicate time periods (left). Quantification of the NRF2 band intensity by ImageJ in the control and cordycepin-treated fibroblasts in the presence of 100 mg/mL CHX for the indicate time periods. NRF2 levels in the untreated cells were normalized to 1(right). **c** Western blot analysis of KEAP1 levels from cordycepin-treated fibroblasts for the indicated times (top). Densitometry quantified data of Keap1 to β-Actin expression ratios, represented as a fold change to non-stimulated cells (bottom). This test was repeated three times. Representative images were shown. **d** Representative immunofluorescence pictures of KEAP1 of the tongues from normal mice and irradiated mice (control, prevention, treatment) 10 days after irradiation dose. **e–g** Fibroblasts were incubated in the absence or presence of MG132 (10 μM), (CQ, 20 μM) or 3-methyladenine (3MA, 5 mM) for 4 h in advance and then were treated with 200 μM cordycepin for the indicated times (kept with MG132, CQ or 3MA), and total proteins were harvested for detection of NRF2, KEAP1 and LC3 by western blot. **h** Western blot analysis of Atg7, p62, LC3 levels from cordycepin-treated fibroblasts for the indicated times. **i** Fibroblasts were treated with 200 μM cordycepin observed for autophagy using transmission electron microscopy. Arrows represent autophagy vacuoles. **j–k** Fibroblasts were transfected with control siRNA (si-Ctl), siRNA against Atg7 (si-Atg7) or siRNA against p62 (si-p62), and then were treated with 200 μM cordycepin for the indicated times, and total proteins were harvested for detection of NRF2, Atg7 or p62, KEAP1 and LC3 by Western blot. Bars represent 50 μm (**d**). Data in **a** and **c** represent the means ± S.D. ($n = 3$, **$P < 0.01$; student's $t$-test; ns not significant)

Asp291 are key residues corresponding for the binding of site2 (Fig. 8i).

To confirm the key role of AMPKγ1/α1 in cordycepin-induced AMPK activation, AMPKγ1 and AMPKα1 specific siRNA were developed. Interference with AMPKγ1-specific siRNA does not affect the expression levels of AMPKα1/2 subunit at the mRNA level (Supplementary Fig. 10g), nor does it influence the transcription of the other two AMPKγ isoform (AMPKγ2/γ3; Supplementary Fig. 10g). Similarly, interference with AMPKα1-specific siRNA did not affect the expression levels of AMPKγ1/2/3 subunits and the transcription of AMPKα2 (Supplementary Fig. 10g). However, the stimulating effect of cordycepin on the phosphorylation of AMPK was dramatically decreased when AMPKγ1 or AMPKα1 expression was sufficiently inhibited (Fig. 8j). Therefore, the interaction with the AMPK γ1/α1 subunit could be the key mechanism of cordycepin-mediated AMPK activation.

In order to verify the generality of the AMPK-autophagy-Keap1-NRF2 pathway in mitigating radiation ulcer, we applied two kinds of AMPK agonists (AICAR and metformin) in skin ulcer model. Both AICAR and metformin showed increased AMPK phosphorylation at Thr172 (p-AMPK) (Supplementary Fig. 11a), but their mitigating effects varies. The AMP-mimic AICAR, was able to prevent skin ulcer slightly, but was not as good as cordycepin (Supplementary Fig. 11b), while metformin failed to prevent skin ulcer. These differences may be explained by the different mechanisms of AICAR and metformin to activate AMPK[40–42].

## Discussion

Radiation ulcer is damage to the skin or other biological tissue caused by exposure to radio frequency energy or ionizing radiation. Being a less studied problem, no precise guideline is present for its management. Meanwhile, the mechanisms of unpredictable and uncontrolled extension of the radiation ulcer are not yet completely understood[4]. Ionizing radiation (IR) can generate a high level of DNA damage which often leads to the induction of apoptosis and cellular senescence in vitro and in vivo[7,8,43]. Induction of apoptosis in rapidly regenerating tissues is believed to be responsible for short-term side effects such as myelosuppression. However, the accumulation of dysfunctional senescent cells in tissues combined with a reduction in the proliferative potential of progenitor/stem cells account for the development of long-term complications[9]. Meanwhile, through secretion of the senescence-associated secretory phenotype (SASP), a broad repertoire of cytokines, chemokines, matrix remodeling proteases and growth factors, senescent cells paracrinely promote tissue dysfunction and deterioration[18]. Senescent cells (SCs) accumulate with age and after genotoxic stress, such as total-body irradiation

(TBI)[16,44]. Here, we provide the first evidence that persistent DNA damage foci forms and senescent cells accumulate in radiation ulcer, and we found senescent cells accelerated the development of radiation ulcer. This led us to hypothesize that preventing cell senescence may represent a prospective strategy to mitigate radiation ulcer.

Cordycepin, a natural nucleoside analogue isolated from culture broth of Cordyceps militaris, has been shown to have a variety of biological functions, including anti-tumor, antiviral, anti-oxidant, and anti-inflammatory activities[45]. Importantly, cordycepin has been shown to attenuate age-related oxidative stress and enhances antioxidant capacity in rats[46]. It has also been shown to prevent rat hearts from ischemia/reperfusion injury partially by activating antioxidant defense[47,48]. Cordycepin represents a promising agent for future potential clinical application. In this study, we identified that cordycepin can mitigate the radiation-induced skin, intestine ulcer and mucositis effectively by preventing DNA damage and cell senescence, it is expected to help the patients suffer from radiation ulcer.

NRF2 is a major stress responder that transcriptionally activates antioxidant and cytoprotective genes through binding to ARE motifs[49]. Focusing on the NRF2 pathway is a suitable strategy to delay the senescence process[50,51] and activation of NRF2 are useful for mitigation of gastroduodenal ulcers[52,53]. Here, we observed that cordycepin prevents radiation-induced cell senescence and SASP are NRF2-dependent. These beneficial effects are achieved by promoting p62-dependent autophagic degradation of Keap1, a cysteine-rich protein that acts as a substrate adaptor for the ubiquitination of NRF2 by the Cul3-Rbx1 E3 ubiquitin ligase complex[33,34]. On the other hand, prolonged activation of NRF2 promotes fibroblast senescence as reported by Hiebert et al.[54], these apparent differences likely reflect the specific context, including cell type, stresses to which these cells are exposed and the exact level of NRF2 activation/inhibition achieved. Phosphorylation of AMPK causes inactivation of mTOR and then induces autophagy[36]. In a recent study, Cordycepin was identified to activate AMPK via interaction with the γ1 subunit only in lipid regulation of HepG2 cancer cells[55], here, in our study, we further identified cordycepin can activate AMPK through the interaction with both the α1 and γ1 subunit of AMPK near the autoinhibitory domain which directly relieve autoinhibition, then to promote autophagic degradation of Keap1, and this mechanism is demonstrated to play an important role in the prevention of cell senescence and mitigation of radiation ulcer. Meanwhile, AMPK affects cellular redox state by upregulating antioxidant enzymes[56]. However, the mechanisms by which AMPK regulates antioxidant responses remain largely unknown. Our study demonstrates that activating NRF2 represent a mechanism for AMPK to regulate antioxidant responses

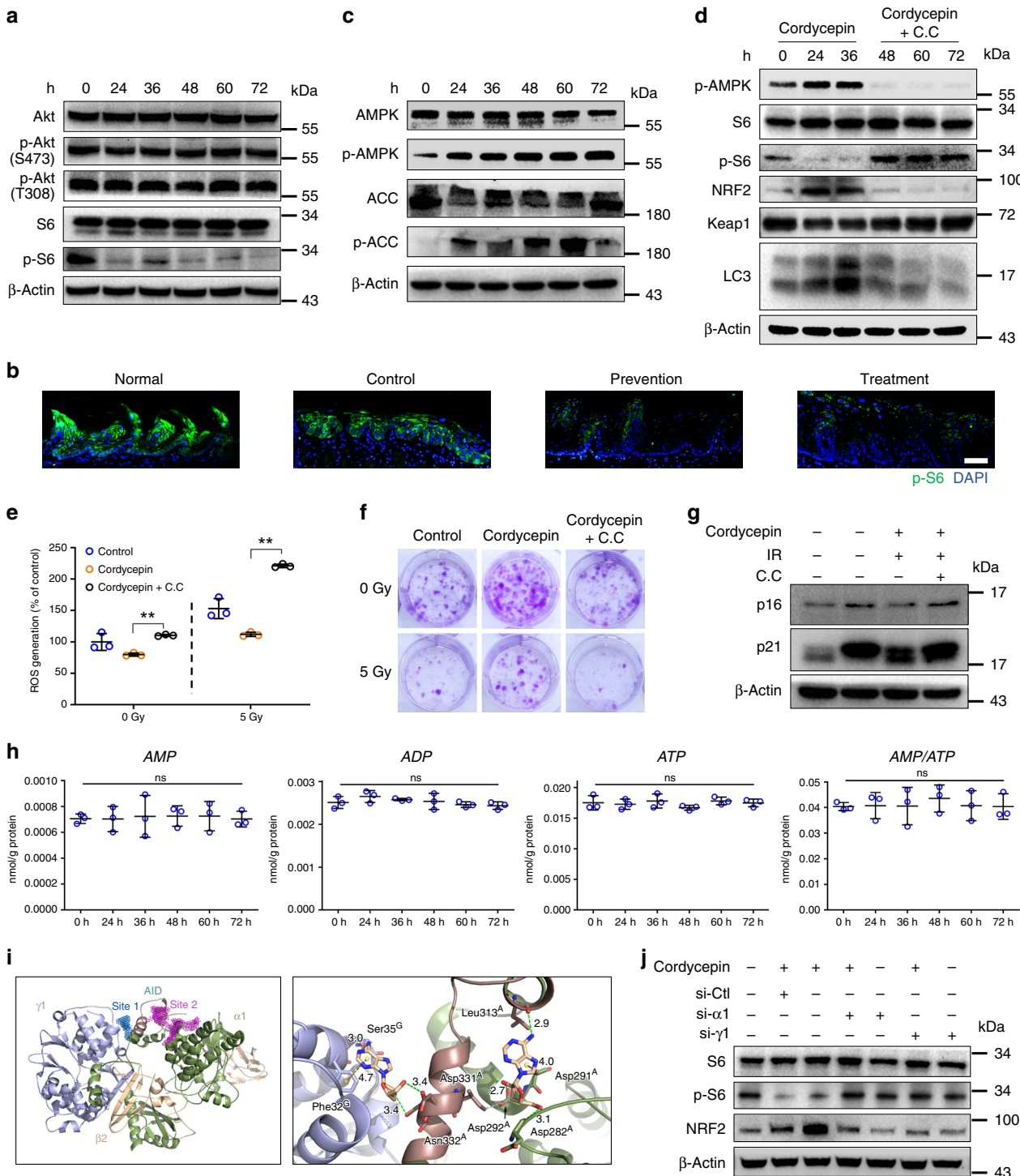

and activation of AMPK or NRF2 may thus represent therapeutic targets for preventing cell senescence.

It has been demonstrated that despite different compounds activating the same target, the effect maybe varies because of the nature and timing of the activation[57]. In our study, the AMP-mimic AICAR, was able to prevent skin ulcer slightly, but was not as good as cordycepin, while metformin failed to prevent skin ulcer. The activation of AMPK by metformin is believed to be an indirect consequence of the metabolic stress induced by inhibition of oxidative phosphorylation and the resulting increase in cytoplasmic AMP levels[42]. In contrast, ZMP, the active form of AICAR can activate AMPK directly by binding to the γ and β subunits, respectively, and are not dependent on the energetic and metabolic status of the cell[42]. Cordycepin can promote AMPK activity directly which is not because of stimulation of upstream kinases, or by altering the AMP/ATP ratio. Together, these observations indicate that the role of AMPK in radiation ulcer is complex and likely context dependent.

In summary, our findings provide an avenue to mitigate radiation ulcer by preventing cell senescence and cordycepin as a natural NRF2 activator by directly binding with the α1 and γ1 subunit of AMPK near the autoinhibitory domain which directly relieve autoinhibition and can be a promising radiation ulcer mitigator via blocking cell senescence (Fig. 9).

**Fig. 8** Cordycepin interacts with α1 and γ1 subunit of AMPK. **a** Akt, p-Akt, S6, p-S6 expression levels from 200 μM cordycepin-treated fibroblasts for the indicated times. **b** Representative immunofluorescence pictures of p-S6 of the tongues from normal mice and irradiated mice (control, prevention, treatment) 10 days after irradiation dose. **c** AMPK, p-AMPK, ACC, p-ACC expression levels from 200 μM cordycepin-treated fibroblasts for the indicated times. This test was repeated three times. **d** Fibroblasts were incubated in the absence or presence of 40 μM Compound C (C.C) for 4 h in advance and then were treated with 200 μM cordycepin for the indicated times (kept with C.C), and total proteins were harvested for detection of p-AMPK, S6, p-S6, NRF2, KEAP1, and LC3 by western blot. **e** Analysis of reactive oxygen species (ROS) levels 24 h after radiation in fibroblasts, fibroblasts were incubated in the absence or presence of 40 μM C.C for 4 h in advance and then were treated with 200 μM cordycepin for 3 days (kept with C.C) before radiation. **f** Representative images of fibroblasts colonies generated in survival assays, fibroblasts were incubated in the absence or presence of 40 μM C.C for 4 h in advance and then were treated with 200 μM cordycepin for 3 days (kept with C.C) before radiation. **g** p16 and p21 expression levels from fibroblast 7 days after radiation, fibroblasts were incubated in the absence or presence of 40 μM C.C for 4 h in advance and then were treated with 200 μM cordycepin for 3 days (kept with C.C) before irradiation. **h** Effect of cordycepin on intracellular AMP, ADP, ATP levels, and the AMP/ATP ratio, fibroblasts were treated with 200 μM cordycepin for the indicated times. **i** Molecular docking study of the binding affinity of cordycepin to AMPK. **j** Fibroblasts were transfected with control siRNA (si-Ctl), siRNA against AMPK-α1 (si-α1) or siRNA against AMPK-γ1 (si-γ1), and then were treated with 200 μM cordycepin for 3 days, and total proteins were harvested for detection of S6, p-S6 and NRF2 by western blot. Bars represent 50 μm (**b**). Data in **e** and **h** represent the means ± S.D. ($n = 3$, **$P < 0.01$; student's $t$-test; ns not significant)

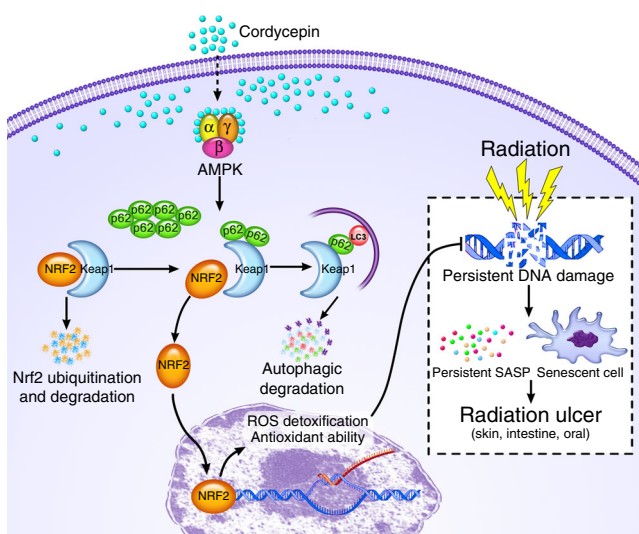

**Fig. 9** Proposed model of cordycepin as radiation ulcer mitigator by preventing cell senescence. Persistent DNA damage foci forms and senescent cells accumulate in radiation ulcer, which involved in the development of radiation ulcer. Cordycepin activates AMPK by directly binding to AMPK protein via interacting with the α1 and γ1 subunit near the autoinhibitory domain (AID), activated AMPK promotes p62-dependent autophagic degradation of Keap1, thus, NRF2 dissociates from Keap1 and moves to the nucleus, where it transactivates ARE-containing antioxidant genes

## Methods

**Source of cordycepin**. The cordycepin was either isolated from culture broth of a Cordyceps militaris[58] or purchased from commercially available source (MCE, Cat. No. HY-N0262). For isolation in brief, the freeze-dried filtrate of the culture broth was redissolved in distilled water to produce cordycepin solution; The solution was filtered through a membrane filter, and the supernatant was left in the refrigerator 5 °C overnight; The crystallized cordycepin was recovered with a filter paper, and the unwanted components were removed by rinse with chilled distilled water; The crystallized cordycepin was freeze-dried. The chemical structures were confirmed by nuclear magnetic resonance (NMR) spectra, including [1]H NMR (Supplementary Fig. 13), [13]C NMR (Supplementary Fig. 14). MOLDI-TOF MS (Supplementary Fig. 15) was performed for further structural identification, sample solution was mixed with an equal volume of DHB (200 mM, 70% ACN/H$_2$O) to form binary matrices, a 2 μL aliquot of the resulting mixtures was added to the MALDI plate, and the mixture was detected after the solvent drying.

**Cell culture**. Primary human fibroblasts were obtained after circumcision of foreskins and approval of the protocol by the ethics committee of Third Military Medical University (Army Medical University). In brief, foreskin tissues were cut into 1–2 cm$^2$ pieces after subcutaneous tissue removal and digested for 1 h at 37 °C in a digestion medium containing 1 mg/ml dispase (Roche). The epidermis was then stripped and digested for another 1 h at 37 °C in the digestion medium

consisting of DMEM with 0.25% collagenase I. The digested cells were then passed through a 75-μm cell strainer, centrifuged, and re-suspended in Iscove's modified Dulbecco's Medium (HyClone) supplemented with 10% fetal bovine serum (Gibco) and 1% streptomycin/penicillin. Cells were grown to >90% confluence, trypsinized (trypsin-EDTA 0.5% w/v, Hyclone), and re-plated for experiments.

Human immortalized keratinocytes (HaCaT cells) were purchased from ATCC (CRL-2310) and maintained in RPMI 1640 growth medium (Gibco) supplemented with 10% fetal bovine serum and 1% streptomycin/penicillin. Cells were grown in a 5% CO$_2$ atmosphere at 37 °C.

**Cell toxicity and proliferation**. The effect of cordycepin on the proliferation of skin fibroblasts and HaCaT cells was assessed using a cell viability assay (Cell Counting Kit-8 (CCK-8; Dojindo Molecular Technologies, Inc., Japan). Briefly, skin fibroblasts and HaCaT cells were seeded in 96-well plates at an initial density of $5 \times 10^3$ cells/well and cultured in different concentrations of cordycepin (100, 200, and 300 μM) for 24, 48, 72 h. Then, 10 μl of CCK-8 solution and 90 μl of culture medium were added to each well at each time point and incubated for 2 h at 37 °C. The absorbance of the samples was measured at 450 nm. The cell toxicity and proliferation data are shown in Supplementary Fig. 16a.

**Apoptosis analysis**. The extent of apoptosis was determined by flow cytometry using an Annexin V/PI staining kit (BD Biosciences), according to the manufacturer's instructions. The cells were washed twice with cold PBS, resuspended in binding buffer, and incubated with annexin-V and PI for 15 min at room temperature in the dark. After incubation, cells were analyzed using flow cytometry.

**Clonogenic survival assay**. Cells were plated in 3.5 cm dishes and treated with vehicle or 200 μM cordycepin for 3 days, and then X-irradiated with 0, 5 or 8 Gy, and kept with vehicle or cordycepin (200 μM) for 24 h. Afterwards, cells were reseeded into six-well plates ($1 \times 10^3$ cells/well) and cultured for up to 14 days until colonies were clearly visible. Colonies were fixed with 4% paraformaldehyde, stained with 0.5% crystal violet, and the colonies which cell number more than 50 were counted. Three independent experiments were performed for each assay. The optimal dosage of cordycepin in cell was determined by clonogenic survival assay (Supplementary Fig. 16b).

**Animal irradiation**. Male Sprague-Dawley (SD) rats and C57BL/6J mice (6–8 weeks) were purchased from Laboratory Animal Center of Third Military Medical University (Army Medical University, AMU). In vivo experiments were conducted in accordance with the Guidelines for the Care and Use of Laboratory Animals of the AMU, and all procedures were approved by the Animal Care and Use Committee of the AMU. To study the effect of cordycepin on radiation induced cutaneous ulcer, the SD rats were divided into four groups randomly (normal, control, prevention and treatment group). The dosage of cordycepin (60 mg/kg) in animal models refer to the literatures[46,48,59–61]. For prevention/treatment group, SD rats were injected intraperitoneally every day for 7 days with cordycepin (60 mg/kg) before/after radiation. Rat right posterior limb (other parts covered with lead board) was exposed to 40 Gy radiation under anesthesia using an X-RAD 160-225 instrument (Precision X-Ray, Branford, CT; filter: 2 mm AI; 50 cm, 300 kV/s, 4 mA, 0.9 Gy/min). To verify the effect of cordycepin on radiation induced intestine injury and mucositis, Abdominal or head and neck area (other parts covered with lead board) was exposed to 12 or 15 Gy irradiation under anesthesia. For prevention/treatment group, C57BL/6 J mice were injected intraperitoneally every day for 7 days (3 days for treatment group in intestine injury) with cordycepin (60 mg/kg) before/after radiation. ML385 (30 mg/kg) and compound C (20 mg/kg) were injected intraperitoneally 3 h before cordycepin injection. AICAR (300 mg/kg) and Metformin (100 mg/kg) were injected

intraperitoneally 24 h before radiation and injected every day for 7 days. The schematic of the animal experimental strategy is show in Supplementary Fig. 17.

**The transplantation of senescent cells**. Full-thickness skin samples were obtained from the dorsum of neonatal SD rats, digested at 4 °C overnight with 0.25% trypsin (Hyclone, China). Next, the dermis layer was dissociated by flushing with D-Hank's solution, and the suspension was filtered through a nylon mesh to remove cellular debris and centrifuged. The cells were received 8 Gy irradiation after adhered to the wall, 7 days after radiation, $2 \times 10^7$ senescent cells (100 µl) mixed with an equal volume of growth factor-reduced Matrigel (BD Biosciences) were subcutaneously injected into the legs after 40 Gy radiation.

**SA-β-galactosidase staining**. Cells were plated in 3.5 cm dishes and and treated with vehicle or 200 µM cordycepin for 3 days. Cells were then X-irradiated or not, and further treated with vehicle or cordycepin(200 µM) for 24 h. Then, cells were passed and assessed 7 days after plating. SA–β-gal staining was done using a SA-β-gal staining kit (Cell Signaling) according to the manufacturer's instructions. Cells were fixed with 4% paraformaldehyde for 5 min, washed and incubated at 37 °C with fresh SA-β-gal solution (PH = 6.0), staining was evident after 24 h. Senescent cells were identified as blue-stained cells under light microscopy, a total of 1000 cells were counted in six random field to determine the percentage of SA-β-gal$^+$ cells. For SA-β-Gal activity in the skin, 5-mm frozen sections were fixed with 0.5% glutaraldehyde for 15 min, washed with PBS and were incubated in SA-β-Gal staining solution (PH = 6.0) for 18 h at 37 °C. Then the nucleus was stained with hematoxylin.

**Immunofluorescence staining**. Cells were cultured on glass coverslips and treated with vehicle or 200 µM cordycepin for 3 days prior to irradiation (5 Gy). Cells were fixed at specific time point post-radiation, permeabilized with 0.1% Triton-X 100 and non-specific binding was blocked with 10% FBS in PBS for 1 h. Fixed cells were incubated with the primary antibody overnight at 4 °C, followed by 1.5 h with the secondary antibody. Then nuclei were stained with DAPI. Images were captured using fluorescent microscope (Olympus BX51). To count the γ-H2AX per nucleus, in each experiment, 50 cells per time point (7 random field) were counted randomly to quantify the number of γ-H2AX per nucleus and all images shown are representative of one of three independent experiments. For tissue immunofluorescence, cryosections were fixed with 4% paraformaldehyde in PBS. After washing three times with PBS, cells were permeabilized with 0.1% Triton-X 100 and non-specific binding was blocked with 10% FBS in PBS for 1 h. Slides were then incubated with the primary antibody overnight at 4 °C, followed by a 1.5 h incubation with the secondary antibody. Then nuclei were stained with DAPI. Images were captured using fluorescent microscope (Olympus BX51). The Ki-67 of skin tissues was imaged by using a laser confocal scanning microscope. Values correspond to the average of at least 6 different pictures each from three different mice or rat. Primary antibodies used were γ-H2AX, pS6, Keap1 (1:200, Cell Signaling), BrdU (1:200, Biolegend), NRF2, Ki-67, p63, Cytokeratin14, FOXO3a (1:200, Abcam). Secondary antibodies to different species IgG were Alexa Fluor® 594 (red) or 488 (green) conjugated (1:500 for all, Invitrogen).

**Histological examination**. Rat skin, tongue tissues and intestine were fixed, embedded in paraffin, cut in 3 µm sections, and stained with H&E.

**Crypt microcolony assay**. The crypt microcolony assay was used to quantify stem cell survival by counting regenerated crypts in H&E-stained cross-sections 4 days post irradiation[62,63]. Surviving crypts were defined as containing five or more adjacent chromophilic non-Paneth cells, at least one Paneth cell and a lumen. The number of surviving crypts was counted in 3 circumferences in each group.

**Enzyme-linked immunosorbent assay (ELISA)**. The concentrations of mouse/rat inflammatory cytokines from plasma samples were measured with IL1β, IL6 and TNFαELISA Kits from ProteinTech according to the manufacturer's protocols.

**Determination of ATP or ADP and AMP content**. ATP content was determined by an Enhanced ATP Assay Kit (Beyotime), ADP and AMP contents were detected by ELISA-based assays (Mlbio) according to the manufacturer's instructions.

**Quantitative real-time PCR analysis**. Total RNA was extracted using RNAiso Plus (TaKaRa/Clontech, Mountain View, CA, USA). cDNA synthesis was done following the manufacturer protocol (Maxima First Strand cDNA Synthesis Kit, Thermo Scientific, K1671). Real-time PCR was performed using a SYBR Green qPCR master mix (Takara) according to the manufacturer's protocol. The primers for the qRT-PCR are listed in supplementary table 1. All data were normalized to the control using Actin or GAPDH as internal control.

**Analysis of ROS**. Fluorescent dye 2′,7′-dichlorofluorescin diacetate (H2DCF-DA; Beyotime) and MitoSOX Red mitochondrial superoxide indicator (Molecular Probes) were used to measure the intracellular ROS and Mitochondrial superoxide.

In brief, Cells were harvested at a density of $5–10 \times 10^5$/mL, incubated with 5 µM H2DCF-DA or MitoSOX Red for 20 min at 37 °C, then oxidation of H2DCF and MitoSOX Red were detected by flow cytometry. Three independent experiments were performed in triplicate for each group.

**Western blot analysis**. Total proteins were extracted using ice-cold RIPA buffer containing protease inhibitor cocktail (Roche). Protein extraction kit (Beyotime) was used to extract nuclear and cytoplasmic protein. The concentrations were determined using a BCA kit (Beyotime). Equal amount of protein from each sample was run in 8–12% Tris-glycine SDS-PAGE gel, followed by transfer to PVDF membrane (Millipore). Membranes were blocked with 5% skim milk and probed with primary antibodies at 4 °C overnight, followed by incubation with horseradish peroxidase-conjugated anti-IgG (Beyotime) for 1 h at room temperature. The intensity of bands was visualized and determined using an enhanced chemiluminescence detection system (Bio-Rad Laboratories). Primary antibodies used were: γ-H2AX, GPX-1, Akt, p-Akt, p21, S6, p-S6, p-LKB1, LC3B, AMPK, p-AMPK, Atg7, p62, ACC, p-ACC (1:1000, Cell Signaling), SOD1, SOD2, Catalase, p-NRF2, FOXO3a, p-FOXO3a (1:1000, Abcam), p16, NRF2, Lamin B1, GCLC, GCLM, HMOX1, NQO1, TXN, Keap1, SIRT1, SIRT3, SIRT6, LKB1, CAMKK2 (1:1000, ProteinTech), Actin and GAPDH (1:1000, Beyotime). The primary images (Supplementary Fig. 18) were cropped for presentation.

**RNA interference and plasmid transfection**. After cells were grown to 30–50% confluence, cells cultured in Opti-MEM Reduced Serum Media (Invitrogen) were transfected with siRNA oligomers (50 nM) human small interfering RNAs (siRNA, GenePharma), negative control RNA (NC siRNA) or Keap1 plasmid (Gene-Pharma) with Lipofectamine 3000 (Invitrogen) according to manufacturer's instructions. Fresh culture medium was added to cultures 6 h post-transfection. Transfected cells for other experiments were performed 24 h post-transfection. The siRNAs for RNA interference are listed in supplementary table 2.

**NRF2 transcriptional activity assay**. The transcriptional activity of NRF2 was determined using a luciferase-based transcription activation assay. A vector carrying a NRF2 promoter-controlled luciferase gene (firefly luciferase) and a vector carrying the control luciferase (Renilla luciferase) from an ARE reporter kit (BPS Bioscience) was transiently co-transfected into the cells using Lipofectamine 3000 (Invitrogen). After transduction for 24 h, the cells were treated with cordycepin before being subjected to the luciferase assay with the Dual-Glo Luciferase Assay System (Promega). Briefly, the cells were incubated with firefly luciferase substrate for 10 min prior to measuring luminescence in a 96-well luminescence plate reader. Subsequently, Renilla luciferase was measured after the addition of Dual-Glo Stop & Glo reagent into the wells with a 10-min incubation. The ratio of luminescence from firefly and Renilla was calculated to normalize and compare NRF2 transcriptional activity.

**Transmission electron microscopy**. Cells were harvested and immediately fixed in 2.5% glutaraldehyde overnight at 4 °C and post-fixed with 2% osmium tetroxide for 1 hour at 37 °C. Subsequently, cells were embedded and stained using uranyl-lacetate/lead citrate. The sections were imaged using a TEM (JEM-1400PLUS, Japan)

**Molecular docking**. The X-ray structure of AMPK α1-β2-γ1 trimer was downloaded from RCSB Protein Data Bank (PDB code: 4RER) as the receptor structure. UCSF Chimera[64] was used to remove all but α1-β2-γ1 amino acids (chain A, B, and G). Hydrogens were added and AMBER ff14SB force field assigned by using the Dock Prep module. The 3D structure of ligand Cordycepin was obtained from PubChem and prepared by energy minimization with AM1-BCC[65] charges assigned in Chimera. The DMS tool was employed to generate the molecular surface of receptor using a probe atom with a 1.4 Å radius. An online tool POCASA was used to predict potential binding sites. The sphgen module was then used to generate spheres filling the sites, and the Grid module generate grid files for energy evaluations. DOCK 6.7[66,67] program was utilized to conduct semi-flexible docking where 10000 different orientations were produced. van der Waals and electrostatic interactions between protein and ligand poses were calculated. After that, clustering analysis were performed (RMSD threshold was set 2.0 Å) to obtain the best scored poses.

**Statistical analysis**. All data are presented as means ± standard deviations. Statistical analyses were applied using the Student's $t$-test and one-way analysis of variance to determine statistical significance. Asterisks denote statistical significance (*$P < 0.05$; **$P < 0.01$). Statistical analyses were carried out using the the SPSS 13.0 package (SPSS Inc., Chicago, IL, USA).

**Reporting summary**. Further information on research design is available in the Nature Research Reporting Summary linked to this article.

## Data availability

The authors declare that all data supporting the findings of this study are available within the paper and its Supplementary Information files, and from the authors on request. The source data underlying Figs. 2c, e, i, 3d, 5a–d, 6f, i, l, 7a, c, 8e, and h and Supplementary Figs. 1b, d, e, 2c, e, f, h, 3b, c, 4a–c, 5b–f, 6c, d, 7c, h–j, 9b–e, h, 10d, g, and 16a are provided as a Source Data file.

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

## Acknowledgements

We thank Qing Zhou and Liao Wu for the immunofluorescence. This work was supported by the National Key Research and Development Program (2016YFC1000805), and University Innovation Team Building Program of Chongqing (CXTDG201602020), and intramural research project grants (AWS17J007, and 2018-JCJQ-ZQ-001).

## Author contributions

C.S. and Z.W. designed, carried out and analyzed data from most of the experiments and wrote the manuscript with input from all co-authors; C.S. and X.F. conceived and supervised the study. Z.C., Z.J., P.L., Lang. L., Y.H., H.W., Yu. W., and Lei. L. performed experiments; X.T., D.L, T.J., YW. W., Yang. W., F.L., C.Z., L.C., Y.G., Y.L., F.Y., C.H., H.M., J.C., T.C. analyzed and interpreted data from experiments; all authors discussed the results and commented on the manuscript.

## Additional information

**Competing interests:** The authors declare no competing interests.

