## [Peer Review File · Nature Communications]

Reviewers' comments:

Reviewer #1 (Remarks to the Author):

In this manuscript, the authors report that persistent DNA damage foci along with cell senescence are two key factors that contribute to radiation-induced ulcer development. They also tested cordycepin, a natural nucleoside analogue isolated from a culture broth of *Cordyceps militaris*, as a novel potent mitigator to inhibit radiation-induced senescence and ultimately the ulcer development. They clearly demonstrated that cordycepin treatment caused an increase of NRF2 levels through p62-dependent autophagic degradation of Keap1. Furthermore, to explore the potential mechanisms they found that NRF2 binds directly to α 1 and γ 1 subunits of AMPK near the autoinhibitory domain, then promote autophagic degradation of Keap1 and increase nuclear NRF2 activity where transactivates ARE-containing antioxidant genes.

The manuscript contains several novel findings and for the most part, the results support the main conclusions. However, I have several concerns regarding the experimental design and data interpretation:

Major points:

1. The authors studied the radiation-induced ulcer on the posterior limb from SD rats and the intestine and tongues from C57BL/6J mice. It will be more clear to verify in each figure legend and in the respective portions of the manuscript the species/origin of these tissues.
2. They monitored the rats for 60 days and the mice for 4 and 9 days post-RT depending on the tissue. Do the authors have survival curves of mice especially those that have been irradiated in the abdominal area with 12Gy? This is an important consideration, since any clinically effective protector/mitigator needs to demonstrate reduction in RT-induced death. Also, why was the dose of 12Gy used for radiation of the intestine? Please clarify.
3. Figure 1d, the γ -H2AX levels on unirradiated skin tissues appear quite high (especially in the second rat). Also, were the levels of γ -H2AX checked at earlier time points, i.e. 1h post-RT to show that indeed γ -H2AX is induced as a marker of double-strand DNA damage?
4. Figure 1g, it is well known that the levels of IL1b, IL6, and TNF α are affected not only as part of the SASP but also as part of the inflammatory response in acute radiation injury. Due to the high dose in both skin and intestine-which also causes systemic effects-have authors checked the circulating levels of IL1b, IL6 and, TNF α in blood?
5. In Figure 3, the γ -H2AX levels with or without cordycepin treatment in different tissues are shown. It is proposed that cordycepin treatment prevents the radiation-induced DNA damage and cell senescence in vivo. Persistent DNA damage is often also accompanied by cell death, so the authors also need to assay the levels of apoptosis (i.e. TUNEL assay or cleaved-caspase3 staining). They also need to present the SA-b-gal staining on other irradiated tissues as well (intestine, tongue). Moreover, in panel 3d, it appears that the skin on both the prevention and treatment setting is quite thickened and most likely probably fibrotic. It will be interesting to perform Masson's Trichrome staining on these tissues to analyze the level of collagen deposition.
6. Figure 5b, they need to add the 0h time point to both conditions as a control. Also, they need to assay for the levels of apoptosis and the proliferation status post-IR, with and without cordycepin treatment.
7. In Figure 5d-g, why did the authors choose to analyze the fibroblasts at 7-days post-IR and not another time point? Please clarify.

8. Figure 8b, has p-S6 staining been done on the skin tissues which is the most studied tissue in this study?

9. The authors showed that the ROS levels were high after IR and were reduced significantly after cordycepin treatment. In Figure 8e they showed that Compound C is able to reverse this phenotype and the ROS levels were significantly increased compared to cordycepin treatment indicating that AMPK is involved in the mechanism. Demonstration of ROS involvement would be bolstered by use the N-acetyl-L-cysteine (NAC) as a standard ROS inhibitor to compare its effectiveness with that of cordycepin. They also need to use the NAC as a control in some of the in vitro experiments, i.e. the SA-b-gal staining.

10. The authors have successfully reversed the radiation ulcer development after cordycepin treatment by blocking the persistent DNA damage foci and cell senescence. This means that the treated tissues will continue to proliferate even if they have very low levels of persisted DNA damage. One of the long-term effects post-IR is the collagen deposition and development of fibrosis. This phenomenon is mostly initiated by the persisted inflammation in the damaged area. How do the authors believe that the cordycepin treatment will impact the long-term effects of IR and if they have results to show that the irradiated mice on the intestine survived and showed protection from the fibrosis.

Minor issues:

1. A clear description of the method used to quantify the surviving crypts and which marker was used to analyze that needs to be added in Materials & Methods.
2. Line 84, replace the "µm" with "µM".
3. Figure 5a, there should be description in the figure legend the IR dose used for the in vitro setting.
4. Figure 7c, add a quantification graph or the -fold change under the WB for KEAP1.

Reviewer #2 (Remarks to the Author):

In this paper, the authors identified cordycepin as a modulator of AMPK activity, which ultimately results in the autophagy-dependent degradation of Keap1, and stabilization of the transcription factor Nrf2. The Nrf2-dependent transcription program is then able to protect cells from incurring damage in response to irradiation, thus minimizing the induction of cellular senescence. The authors suggest that this pathway could provide a novel therapeutic target for the prevention of ulcer formation in response to radiation treatment performed during bone marrow transplantations or cancer therapy. While this manuscript contains some interesting ideas, convincing lines of data and satisfactory explanations of interpretations are not provided. In addition, several key experiments are missing or have not been performed. This reviewer thinks that more work is required before this paper could be considered for publication in Nature Communications.

Major points

1. In this study, the mechanism how cordycepin activate Nrf2 is attributed to the Keap1 degradation through the AMPK-autophagy activation. However, reduction of Keap1 protein level seems to be marginal (Figure 7c, e, f, g, j, and k), and presented evidence is not so convincing. Another hypothesis might be appropriate. For example, there is a report that AMPK directly phosphorylates Nrf2 and promotes nuclear accumulation of Nrf2 (Joo et al, MCB 2016).
2. In order to rigorously support the proposed model, the experiments in Figures 1 to 4 should also be carried out in Nrf2 knockout mice. This line of mouse is commonly used in many experimental settings, and results derived from this knockout model would much more clearly support the requirement of Nrf2 for this process. As AMPK has many effects on the cellular phenotype independent of Nrf2, the use of Nrf2 knockout mice would show that Nrf2 is the downstream factor responsible for

senescence suppression. In addition, as cordycepin is a nucleoside analogue, it may potentially modulate the activity of other kinases, in addition to AMPK, highlighting the need for more robust in vivo mechanistic support for the author's model.

3. Line 55, there is no satisfactory explanation how cordycepin was identified as potential drug for radiation ulcer. Detail of the drug screening should be properly described.

4. The physiological relevance of Ser40 phosphorylation of Nrf2 has not been rigorously demonstrated in the literature, and therefore Nrf2 activity is more commonly shown by immunoblot analysis of non-phosphorylated Nrf2. Do the authors think that Ser40 has some relevance to their study?

5. In Fig 6F, the authors present qRT-PCR data for NRF2 target genes in fibroblasts treated with cordycepin. Do they have corresponding qRT-PCR data from the mouse models, for example, using the skin, intestine and tongue samples from Fig 3E-G?

6. In Figure 7J, Atg7 knockdown reduced Nrf2 protein. This is not consistent with previous reports that Atg7 knockout liver exhibits significant accumulation of Nrf2 (Komatsu et al, 2010). Appropriate explanations are required. In addition, p62 protein level should be examined.

7. There are some reports that Nrf2 activators are useful for mitigation of ulcer (Mahmoud-Awny et al, 2015; Zhang et al, Pharmacol Rep 2017). These reports should be discussed.

8. There is a recent report that Nrf2 activation in fibroblasts promote cellular senescence and SASP (Hiebert et al, Dev Cell 2018). This idea is opposite to this study. This point should be discussed.

Minor point

9. In Figures 2-4, there is no explanation about "Prevention" and "Treatment". These should be defined in the text.

10. Line 124, reference is required for "NRF2 phosphorylation on serine....."

11. The English language could be improved throughout the text.

Reviewer #3 (Remarks to the Author):

This article titled "Cordycepin as radiation ulcer mitigator by preventing cell senescence" reported a novel mitigator of radiation ulcer, which can inhibit cell senescence and radiation ulcer through AMPK-NRF2 pathway. In general, the research is well designed and the pathway is carefully demonstrated. However, there are several points need to be addressed.

General points:

1. The authors claimed that they "demonstrate for the first time that persistent DNA damage foci and cell senescence play a pivotal role in radiation ulcer development". However, their data only revealed that DNA damage occurred and senescent cells accumulated during the development of radiation ulcer, while lacking sufficient evidence to show how DNA damage and cell senescence "significantly contribute to the unpredictable and uncontrolled ulcer extension". In fact, not only cell senescence, but also cell death and the clearance of damaged cells via immune system can be regulated by AMPK. Thus, the authors must provide additional data to support their claim.

2. The authors found cordycepin to be a potent mitigator of radiation ulcer. As a novel mitigator, the author may provide more information about cordycepin. How did they determine the dosage of cordycepin in cell and animal model? Have they done any toxicity test? Are there any comparison to the previously reported radiation mitigator like rapamycin?

(<https://www.sciencedirect.com/science/article/pii/S1934590912003694>)

Specific points:

1. Figure 7g and h show that Atg7, p62, Keap1 and Nrf2 expressions were not affected by cordycepin within 48 hours. How can the authors explain the results?

2. There is no in vivo evidence to demonstrate the involvement of AMPK-Nrf2 pathway axis in Cordycepin-mediate anti-aging effects.

3. The authors should provide more experiment-based evidence to demonstrate the notion that Cordycepin represses AMPKa1 and g1 interaction.

Responses to reviewers:

We thank the reviewers for their thoughtful and positive comments as well as the opportunity to respond. We have added new data to the revised paper and specifically addressed all the concerns raised by the editors and the reviewers. We have marked the changes in red and modified the formats according to the checklist provided by the journal. We hope that the revised manuscript will now be acceptable for publication in Nature Communications.

Our responses to the reviewers' comments are enumerated below:

Reviewer #1:

Major points:

1. The authors studied the radiation-induced ulcer on the posterior limb from SD rats and the intestine and tongues from C57BL/6J mice. It will be more clear to verify in each figure legend and in the respective portions of the manuscript the species/origin of these tissues.

RE: We thank the reviewer for the comment. We have labeled the species/origin of the tissues in each figure legend and in the respective portions of the manuscript as suggested.

2. They monitored the rats for 60 days and the mice for 4 and 9 days post-RT depending on the tissue. Do the authors have survival curves of mice especially those that have been irradiated in the abdominal area with 12Gy? This is an important consideration, since any clinically effective protector/mitigator needs to demonstrate reduction in RT-induced death. Also, why was the dose of 12Gy used for radiation of the intestine? Please clarify.

RE: We thank the reviewer for the comment. As suggested, we have added new data on the survival curves of mice that have been irradiated in the abdominal area with 12Gy in **figure 3e**, all vehicle-treated mice died 5-7 days after radiation, in contrast, the survive rate in prevention and treatment group are 57% and 48% respectively at least 40 days after radiation.

The dose of 12 Gy is the commonly used dose in abdominal irradiation according to published

reports (Brian J. Leibowitz et al, Nat Commun. 2014; Ioannis I. Verginadis et al, Cancer Res. 2016; Gong Wet al, Cell Death Dis. 2016; Zhou X et al, Life Sci. 2015; Huang EY et. Int J Radiat Oncol Biol Phys. 2009). Further, we have verified that the dose of 12Gy could induce intestine ulcer effectively in our experiments, therefore, we used the dose of 12Gy in radiation-induced intestine ulcer model.

3. Figure 1d, the γ -H2AX levels on unirradiated skin tissues appear quite high (especially in the second rat). Also, were the levels of γ -H2AX checked at earlier time points, i.e. 1h post-RT to show that indeed γ -H2AX is induced as a marker of double-strand DNA damage?

RE: We thank the reviewer for the comment. We have repeated the experiment and verified that the expression of γ -H2AX level is relatively low in normal tissues (Fig.1d). As suggested, we checked the levels of γ -H2AX 1h post-radiation, the fraction of cells with γ -H2AX foci increased sharply within 1h after animals were irradiated (Supplementary Fig. 1a-c).

4. Figure 1g, it is well known that the levels of IL1b, IL6, and TNF α are affected not only as part of the SASP but also as part of the inflammatory response in acute radiation injury. Due to the high dose in both skin and intestine-which also causes systemic effects-have authors checked the circulating levels of IL1b, IL6 and TNF α in blood?

RE: We thank the reviewer for the comment. As suggested, we checked the circulating levels of IL1b, IL6 and TNF α in blood of skin and intestine ulcer models, as shown in supplementary figure 5c and 5d, plasma concentrations of IL-1 β , IL-6, and TNF- α were significantly lower in cordycepin-treated rats/mice than controls.

5. In Figure 3, the γ -H2AX levels with or without cordycepin treatment in different tissues are shown. It is proposed that cordycepin treatment prevents the radiation-induced DNA damage and cell senescence in vivo. Persistent DNA damage is often also accompanied by cell death, so the authors also need to assay the levels of apoptosis (i.e. TUNEL assay or cleaved-caspase3 staining). They also need to present the SA-b-gal staining on other irradiated tissues as well (intestine, tongue). Moreover, in panel 3d, it appears that the skin on both the prevention and

treatment setting is quite thickened and most likely probably fibrotic. It will be interesting to perform Masson's Trichrome staining on these tissues to analyze the level of collagen deposition.

RE: We thank the reviewer for the comment. As suggested, we have tested the levels of cell apoptosis in three ulcer models and observed substantial differences between control and cordycepin-treated mice/rats in irradiated skin, intestine and tongue by TdT-mediated dUTP nick end labeling (TUNEL) assays (**Supplementary Fig. 4a-c**). We also tested the SA- β -gal staining in intestine and tongue in **supplementary figure 5a and b**. Moreover, regarding the reviewer's concern on tissue fibrosis, we performed the microscopic analysis of the skin from cordycepin-treated group and the results did not reveal significant fibrosis with trichrome staining compared with control group (**Supplementary Fig. 3a**).

6. Figure 5b, they need to add the 0h time point to both conditions as a control. Also, they need to assay for the levels of apoptosis and the proliferation status post-IR, with and without cordycepin treatment.

RE: We thank the reviewer for the comment. We have added the data on the 0h time point to both conditions as a control in **figure 2b and 2c**. Moreover, as shown in **supplementary figure 2f**, the proliferation of cells is largely decreased following radiation, while cordycepin improved the cell proliferative potential after radiation. In addition, cordycepin prevents radiation-induced apoptosis (**Fig. 2f**). We also used hydrogen peroxide (H₂O₂) to mimic cell damage caused by oxidative stress in vitro. Fibroblasts preconditioned with cordycepin demonstrated better resistance to acute oxidative stress with less apoptosis (**Fig.2g**). In conclusion, cordycepin improves the cell proliferation and reduces cell death after oxidative stress.

7. In Figure 5d-g, why did the authors choose to analyze the fibroblasts at 7-days post-IR and not another time point? Please clarify.

RE: We thank the reviewer for the comment. It has been reported that cells will become fully senescent at 7-days after ionizing radiation according to literatures (Jianhui Chang et al, Nat Med.

2016; Wang Y et al, Int J Radiat Oncol Biol Phys. 2011). In our study, we did not detect significant increase of SA- β -gal activity at early time (before 7 days) after radiation, however, the SA- β -gal activity is obvious at 7 days after radiation, therefore we choose to analyze the fibroblasts at 7-days post-IR.

8. Figure 8b, has p-S6 staining been done on the skin tissues which is the most studied tissue in this study?

RE: We thank the reviewer for the comment. We have performed the p-S6 staining on the skin tissues, as shown in supplementary figure 9b, cordycepin prevented the p-S6 activity in skin tissues.

9. The authors showed that the ROS levels were high after IR and were reduced significantly after cordycepin treatment. In Figure 8e they showed that Compound C is able to reverse this phenotype and the ROS levels were significantly increased compared to cordycepin treatment indicating that AMPK is involved in the mechanism. Demonstration of ROS involvement would be bolstered by use the N-acetyl-L-cysteine (NAC) as a standard ROS inhibitor to compare its effectiveness with that of cordycepin. They also need to use the NAC as a control in some of the in vitro experiments, i.e. the SA-b-gal staining.

RE: We thank the reviewer for the comment. We agree with the reviewer that reactive oxygen species (ROS) can mediate the deleterious effects of radiation and contribute to DNA damage and the activation of senescence pathways in cells. As suggested, we use the NAC as a control in supplementary figure 2b and 2c, and the results showed that treatment with ROS scavenger N-acetylcysteine (NAC) at least partially abolished the radiation-induced cell senescence, further supporting the involvement of ROS in the cell senescence induced by radiation.

10. The authors have successfully reversed the radiation ulcer development after cordycepin treatment by blocking the persistent DNA damage foci and cell senescence. This means that the treated tissues will continue to proliferate even if they have very low levels of persisted DNA damage. One of the long-term effects post-IR is the collagen deposition and development of fibrosis. This phenomenon is mostly initiated by the persisted inflammation in the damaged

area. How do the authors believe that the cordycepin treatment will impact the long-term effects of IR and if they have results to show that the irradiated mice on the intestine survived and showed protection from the fibrosis?

RE: We thank the reviewer for the comment. In order to evaluate the long-term protection from the fibrosis on the intestine, mRNA expression for COL-1A1, COL-3A1 and fibronectin in the intestine tissue in mice survived 28 days post-radiation were detected, and we didn't find significant development of fibrosis as shown in supplementary figure 3b.

Minor issues:

1. A clear description of the method used to quantify the surviving crypts and which marker was used to analyze that needs to be added in Materials & Methods.

RE: We thank the reviewer for the comment. As suggested, a clear description of the method used to quantify the surviving crypts is added in **Methods-Crypt microcolony assay (Line 474)**.

2. Line 84, replace the “ μm ” with “ μM ”.

RE: We have replaced the “ μm ” with “ μM ” (Line 76) in the manuscripts.

3. Figure 5a, there should be description in the figure legend the IR dose used for the in vitro setting.

RE: We thank the reviewer for the comment. We have added the IR dose used for the in vitro setting in figure 2a to 2c.

4. Figure 7c, add a quantification graph or the -fold change under the WB for KEAP1.

RE: As suggested, we have added the fold change under the WB for KEAP1 in figure 7c.

Reviewer #2:

Major points:

1. In this study, the mechanism how cordycepin activate Nrf2 is attributed to the Keap1 degradation through the AMPK-autophagy activation. However, reduction of Keap1 protein level seems to be marginal (Figure 7c, e, f, g, j, and k), and presented evidence is not so convincing. Another hypothesis might be appropriate. For example, there is a report that AMPK directly phosphorylates Nrf2 and promotes nuclear accumulation of Nrf2 (Joo et al, MCB 2016).

RE: We thank the reviewer for the comment. To exclude the possibility that reduction of Keap1 protein level is marginal and further demonstrate that cordycepin activates NRF2 is attributed to the Keap1 degradation, fibroblasts were transfected with the KEAP1 expressing plasmid and we found that the NRF2 protein level was reversed after treated with cordycepin (**Supplementary Fig. 8i**), suggesting that cordycepin activates NRF2 in a keap1-depedent manner.

On the other hand, as mentioned by the reviewer, there is a report that AMPK phosphorylates NRF2 at the Ser550 residue, which in conjunction with AMPK-mediated GSK3- β inhibition, promotes nuclear accumulation of NRF2 for antioxidant response element (ARE)-driven gene transactivation. However, we didn't find significant increase of GSK3- β (Ser9) in our study (**Supplementary Fig. 9c**). If AMPK directly phosphorylates NRF2, the change is in the NRF2 nuclear localization but not in the protein level as described in **Joo** 's report. However, in our study, we observed the increase of total protein level of NRF2. Therefore, we propose that AMPK directly phosphorylates NRF2 is not the major pathway for cordycepin to activate NRF2 in our model.

2. In order to rigorously support the proposed model, the experiments in Figures 1 to 4 should also be carried out in Nrf2 knockout mice. This line of mouse is commonly used in many experimental settings, and results derived from this knockout model would much more clearly support the requirement of Nrf2 for this process. As AMPK has many effects on the cellular

phenotype independent of Nrf2, the use of Nrf2 knockout mice would show that Nrf2 is the downstream factor responsible for senescence suppression. In addition, as cordycepin is a nucleoside analogue, it may potentially modulate the activity of other kinases, in addition to AMPK, highlighting the need for more robust in vivo mechanistic support for the author's model.

RE: We appreciate the reviewer for the constructive comment. Because it will take at least six months or more to build and breed the NRF2 knockout mice, which is difficult for us in the current situation, we applied a newly developed novel strategy using specific NRF2 inhibitor (ML385) to inhibit the activation of NRF2 (Singh A et al, ACS Chem Biol. 2016; Xinnong Liu et al. Oxid Med Cell Longev. 2018). Notably, using the western blotting, we found that ML385 effectively inhibited the NRF2 activity in vivo (Supplementary Fig. 7a), H&E staining showed that the lung, liver, spleen, heart, kidney and intestine were not significantly influenced by ML385 (Supplementary Fig. 10). Interestingly, ML385 significantly blocked the protective effects of cordycepin in radiation induced skin, intestine and tongue ulcers (Supplementary Fig. 7c-d and Supplementary Fig. 3d). In radiation-induced intestine ulcer model, the survival rates in cordycepin with ML385 group were similar to those of the control group (Supplementary Fig. 3e). Therefore, we further verified that cordycepin mitigates the radiation ulcer is NRF2-dependent.

In addition, we used Compound C to inhibit AMPK in vivo (Supplementary Fig. 7b), H&E staining showed the lung, liver, spleen, heart, kidney and intestine were not significantly influenced by Compound C (Supplementary Fig. 10), the protective effects of cordycepin in radiation induced skin, intestine and tongue ulcers were significantly reversed (Supplementary Fig. 7c-d and Supplementary Fig. 3d), the survival rates in cordycepin with Compound C group were like those of the control group in radiation-induced intestine ulcer model (Supplementary Fig. 3e). These evidence supports that AMPK-NRF2 pathway axis is involved in cordycepin-mediate anti-ulcer effects in vivo.

3. Line 55, there is no satisfactory explanation how cordycepin was identified as potential drug for radiation ulcer. Detail of the drug screening should be properly described.

RE: We thank the reviewer for the comment. In our study, we have demonstrated that radiation ulcer

has long lasting effects in the cells with persistent DNA damage foci, which leads to a wide accumulation of senescent cells in tissues (Fig.1b and 1d). Further, we found senescent cells accelerated the development of radiation ulcer (Fig.1f). Therefore, we hypothesized that preventing cell senescence could be a therapeutic strategy to mitigate radiation ulcer and we established a senescence model induced by radiation in vitro to screen the potential small-molecule which can inhibit cell senescence. By screening the candidate small-molecules, we identified cordycepin, a natural nucleoside analogue compound, can reduce levels of the senescence marker effectively as shown in figure 2d-e and supplementary figure2a-e.

4. The physiological relevance of Ser40 phosphorylation of Nrf2 has not been rigorously demonstrated in the literature, and therefore Nrf2 activity is more commonly shown by immunoblot analysis of non-phosphorylated Nrf2. Do the authors think that Ser40 has some relevance to their study?

RE: We thank the reviewer for the comment. Nrf2 activity is usually tested by immunoblot analysis of non-phosphorylated Nrf2 and we also mainly focus on the non-phosphorylated Nrf2 level in our study. It has been reported that the phosphorylation on serine 40 is a regulatory modification required for NRF2 translocation to the nucleus and downstream protein activation (Huang HC et al, J Biol Chem. 2002; Wang X et al, Advanced Functional Materials 2016; Dehghan E et al, Nat Commun. 2017), so we test the Ser40 phosphorylation of NRF2 to further support the activation of NRF2.

5. In Fig 6F, the authors present qRT-PCR data for NRF2 target genes in fibroblasts treated with cordycepin. Do they have corresponding qRT-PCR data from the mouse models, for example, using the skin, intestine and tongue samples from Fig 3E-G?

RE: We thank the reviewer for the comment. As suggested, we have added qRT-PCR data for NRF2 target genes from the rat skin, mouse intestine and tongue samples in supplementary figure 6i to further identify that cordycepin can activate NRF2 target genes in vivo.

6. In Figure 7J, Atg7 knockdown reduced Nrf2 protein. This is not consistent with previous reports that Atg7 knockout liver exhibits significant accumulation of Nrf2 (Komatsu et al,

2010). Appropriate explanations are required. In addition, p62 protein level should be examined.

RE: We thank the reviewer for the comment. In the reference mentioned by the reviewer, a prominent accumulation of NRF2 in the nucleus was observed in livers deficient in Atg7, the authors describe that the excessive p62 leads to a prominent accumulation of NRF2 in the nucleus which is a concomitant phenomenon when autophagy-deficiency causes cellular stress in the liver and negatively affects hepatocyte function. However, they haven't clarified the role of Keap1 in livers deficient in Atg7. In our study, we identified that autophagy can degrade Keap1, then to promote the increase of NRF2 protein in the cell. Further, autophagy is a double-edged sword and autophagy can play an opposite role in different models. In the reference mentioned by reviewer, unlike the liver, aberrant accumulation of p62 did not induce NRF2 expression in brains. The authors hypothesize that the pathologic process associated with autophagic deficiency is cell-type specific. Therefore, we propose that autophagy plays a different role in regulation of NRF2 in skin compared to liver. As suggested, we have examined the p62 protein level and the result is shown in **figure 7j**.

7. There are some reports that Nrf2 activators are useful for mitigation of ulcer (Mahmoud-Awny et al, 2015; Zhang et al, Pharmacol Rep 2017). These reports should be discussed.

RE: We thank the reviewer for providing the reports. We have discussed this work in the **Discussion** section of the revised manuscript (**Line 353**).

8. There is a recent report that Nrf2 activation in fibroblasts promote cellular senescence and SASP (Hiebert et al, Dev Cell 2018). This idea is opposite to this study. This point should be discussed.

RE: We thank the reviewer for providing the report. In the reference mentioned by reviewer, prolonged activation of NRF2 promotes fibroblast senescence, then increase the production of a senescence promoting matrisome and lead to accelerated wound closure. However, in many cases, NRF2 activation delayed the onset of senescence (Jodaret al., J Gerontol A Biol Sci Med Sci. 2011;

Kapeta et al., J. Biol. Chem. 2010; Volonte et al., Mol. Biol. Cell. 2013; Wanget al., Aging Cell. 2017; Yang et al., Antioxid. Redox Signal. 2013). These apparent differences likely reflect the specific context, including cell type, stresses to which these cells are exposed and the exact level of NRF2 activation/inhibition achieved. In our experiments, we only treated fibroblasts for 3 days before radiation, while in the reference mentioned by reviewer, they treated primary human foreskin fibroblasts with tBHQ every 3 days for 2 weeks, this may cause the different level of NRF2 activation as compared with ours.

Further, they also suggest that senescence can be induced via different pathways and is independent of ROS. While in our study, we found that treatment with ROS scavenger N-acetylcysteine (NAC) at least partially abolished the radiation-induced cell senescence (supplementary figure 2b and 2c), therefore, we propose that ROS is involved in the radiation-induced cell senescence.

We also appreciate the reviewer for the thoughtful comments. We have discussed this point in the manuscript (**Line 357**).

Minor issues:

1. In Figures 2-4, there is no explanation about “Prevention” and “Treatment”. These should be defined in the text.

RE: We thank the reviewer for the comment. We have added explanation about “Prevention” and “Treatment” in **Methods- Animal irradiation (Line 416)**, we also added a schematic of the animal experimental strategy in supplementary figure 15.

2. Line 124, reference is required for “NRF2 phosphorylation on serine.....”

RE: We thank the reviewer for the comment. We have added the references in the text that NRF2 phosphorylation on serine 40 is a regulatory modification required for NRF2 translocation to the nucleus and downstream protein activation (**line 176**).

3. The English language could be improved throughout the text.

RE: We thank the reviewer for the comment. The English language has been revised by an expert whose first language is English.

Reviewer #3:

General points:

1. The authors claimed that they "demonstrate for the first time that persistent DNA damage foci and cell senescence play a pivotal role in radiation ulcer development". However, their data only revealed that DNA damage occurred and senescent cells accumulated during the development of radiation ulcer, while lacking sufficient evidence to show how DNA damage and cell senescence "significantly contribute to the unpredictable and uncontrolled ulcer extension". In fact, not only cell senescence, but also cell death and the clearance of damaged cells via immune system can be regulated by AMPK. Thus, the authors must provide additional data to support their claim.

RE: We thank the reviewer for the comment. DNA damage has been described to induce cell senescence (Muñoz-Espín D et al., Nat Rev Mol Cell Biol. 2014; Campisi J et al., Annu Rev Physiol. 2013; Ovadya Y et al., Biogerontology. 2014). In order to further characterized the involvement of senescent cells in the development of radiation ulcer, senescent cells were subcutaneously injected into the irradiated legs. We found senescent cells accelerated the development of radiation ulcer. The irradiated legs transplanted with senescent cells became red and swollen at 4 days while the control until the 8th day. It began to shed hair on the 4th day in the irradiated legs transplanted with senescent cells which is 4 days earlier than control group. And the ulceration of the skin appeared at 8 days in senescent group while the control until the 13th day. These data support that the senescent cells are involved in the development of radiation ulcer. In order to describe more concisely, we have deleted the description "the unpredictable and uncontrolled ulcer extension".

Further, we agree with the reviewer that not only cell senescence, but also cell death and the clearance of damaged cells via immune system can be regulated by AMPK. However, in this study,

we focused on the role of preventing cell senescence, but not clearing damaged/dead cells in the treatment of radiation ulcer. The results show that cordycepin is a potent mitigator for radiation ulcer by preventing cell senescence. We thank the reviewer for the insightful comment and will identify whether cordycepin could clear the senescent cells or damaged cells via immune system in our future studies.

2. The authors found cordycepin to be a potent mitigator of radiation ulcer. As a novel mitigator, the author may provide more information about cordycepin. How did they determine the dosage of cordycepin in cell and animal model? Have they done any toxicity test? Are there any comparison to the previously reported radiation mitigator like rapamycin? (<https://www.sciencedirect.com/science/article/pii/S1934590912003694>)

RE: We thank the reviewer for the comment. In our study, we have demonstrated that radiation has long lasting effects in the cells with persistent DNA damage foci, which leads to wide accumulation of senescent cells in tissues (Fig.1b and 1d). Further, we found senescent cells accelerated the development of radiation ulcer (Fig.1f). Therefore, we hypothesized that preventing cell senescence could be a therapeutic strategy to mitigate radiation ulcer and we established a cell senescence model induced by radiation in vitro to screen the potential small-molecule which can inhibit cell senescence. By screening the candidate small-molecules, we identified cordycepin, a natural nucleoside analogue compound, can reduce levels of the senescence marker effectively as shown in figure 2d-e and supplementary figure 2a-e.

We also have added the cell toxicity and proliferation data in supplementary figure 14a. In supplementary figure 10, H&E staining showed that the lung, liver, spleen, heart, kidney and intestine are not significantly influenced by cordycepin. The optimal dosage of cordycepin in cell was determined by clonogenic survival assay since each large colony is expected to grow from a single surviving self-renewing cell, we found 200 μ M is the optimal concentration for clonogenic survival (Supplementary Fig. 14b). The dosage of cordycepin (60mg/kg) in animal models is refer to the literatures (Ramesh T et al. Exp Gerontol. 2012; Park ES et al. Cardiovasc Toxicol. 2014; Ma L et al. Nutr Res. 2015; Wang F et al. Oncotarget. 2015; Tianzhu Z et al. Inflammation. 2015; Chen M et al. J Ethnopharmacol. 2012).

We appreciate the reviewer for providing the report about the rapamycin. However, in supplementary figure 3d, rapamycin did not protect from mucositis compared with cordycepin when mice were irradiated with high intensity in a single dose, indicating that the protection conferred by cordycepin treatment might be better than rapamycin in a high single dose.

3. Figure 7g and h show that Atg7, p62, Keap1 and Nrf2 expressions were not affected by cordycepin within 48 hours. How can the authors explain the results?

RE: We thank the reviewer for the comment. We have improved the contrast of the picture and repeated the experiments. We also add the fold change under the WB for Atg7, p62, Keap1 and NRF2 of figure 7f-g in supplementary figure 8c, 8d and 8e. The Atg7, p62, Keap1 and NRF2 expressions are affected by cordycepin within 48 hours.

4. There is no in vivo evidence to demonstrate the involvement of AMPK-Nrf2 pathway axis in Cordycepin-mediate anti-aging effects.

RE: We appreciate the reviewer for the consideration that there should have in vivo evidence to demonstrate the involvement of AMPK-NRF2 pathway axis in cordycepin-mediate anti-aging effects. As suggested, we applied a newly developed novel strategy using specific NRF2 inhibitor (ML385) to inhibit the activation of NRF2 in our study (Singh A et al, ACS Chem Biol. 2016; Xinnong Liu et al. Oxid Med Cell Longev. 2018). Notably, using the western blotting, we found that ML385 effectively inhibited the NRF2 activity in vivo (Supplementary Fig. 7a), H&E staining showed that the lung, liver, spleen, heart, kidney and intestine were not significantly influenced by ML385 (Supplementary Fig. 10). Interestingly, ML385 could significantly blocked the protective effects of cordycepin in radiation induced skin, intestine and tongue ulcers (Supplementary Fig. 7c-d and Supplementary Fig. 3d). In radiation-induced intestine ulcer model, the survival rates in cordycepin with ML385 group were similar to those of the control group (Supplementary Fig. 3e). Therefore, we further verified that cordycepin mitigates the radiation ulcer is NRF2-dependent.

Furthermore, we used Compound C to inhibit AMPK in vivo (Supplementary Fig. 7b), H&E staining showed that the lung, liver, spleen, heart, kidney and intestine were not significantly influenced by

Compound C (Supplementary Fig. 10), the protective effects of cordycepin in radiation induced skin, intestine and tongue ulcers were significantly reversed (Supplementary Fig. 7c-d and Supplementary Fig. 3d), the survival rates in cordycepin with Compound C group were like those of the control group in radiation-induced intestine ulcer model (Supplementary Fig. 3e). These evidences further support that AMPK-NRF2 pathway axis involved in cordycepin-mediate anti-ulcer effects in vivo.

5. The authors should provide more experiment-based evidence to demonstrate the notion that Cordycepin represses AMPK α 1 and γ 1 interaction.

RE: We thank the reviewer for the comment. In our study, we demonstrate that cordycepin can interact with the α 1 and γ 1 subunit near the autoinhibitory domain to relieve autoinhibition, but not directly represses AMPK α 1 and γ 1 interaction. First, we excluded the upstream pathway that can activate AMPK, therefore, we hypothesized that cordycepin activates AMPK by directly binding to AMPK protein. Second, the binding ability of cordycepin to each subunit of AMPK was evaluated by molecular docking using POCASA, we found cordycepin binds to α 1 and γ 1 subunit near the autoinhibitory domain with relative high affinity. Third, in order to confirm the key role of AMPK γ 1/ α 1 in cordycepin-induced AMPK activation, AMPK γ 1 and AMPK α 1 specific siRNA were established and the stimulating effect of cordycepin on the phosphorylation of AMPK was dramatically decreased when AMPK γ 1 or AMPK α 1 expression was sufficiently inhibited (Fig. 8j). Therefore, cordycepin interacts with the AMPK γ 1/ α 1 subunit may be the key mechanism of cordycepin-mediated AMPK activation.

Reviewers' comments:

Reviewer #1 (Remarks to the Author):

The authors have responded to all the issues raised by this reviewer. Additional experiments, particularly follow-up on overall survival of mice treated with cordycepin and showing significant increase in overall survival bolster considerably the overall impact. Moreover, addition of important 0 time point controls for proliferation and apoptosis effects of in IR or IR+ cordycepin groups raise confidence for the validity of the in vitro observations. Collectively, the results presented in this manuscript make a strong argument for radiation mitigation effects of cordycepin and have important clinical ramifications.

Reviewer #2 (Remarks to the Author):

Main comments

This manuscript has been revised substantially, but this reviewer feels that this manuscript needs to be improved more in a few points.

The relationship between radiation-induced DNA damage, SASP, AMP Kinase phosphorylation, and Nrf2 activity have not been adequately addressed in the mouse model. In Fig 4a-c, additional panels should be added to show the effect of Nrf2 inhibition (cordycepin plus ML385) and AMP kinase inhibition (cordycepin plus compound C) on DNA damage (γ -H2AX) in response to radiation.

Similarly, Fig 4e-f should also be expanded to include cordycepin plus ML385 and cordycepin plus compound C, in order to confirm the role of AMPK and Nrf2 in cellular senescence (p16 and p21 Western blots) and the SASP phenotype in the mouse model, and not merely in cell lines. This additional data would also clearly show the causality between AMPK, Nrf2, DNA damage and SASP in the mouse radiation model, as this is a critical factor which is currently lacking.

On line 71, the authors claim that they carried out a screen to identify cordycepin, yet no data or details of the screen are given within the manuscript. From the mechanism proposed by the authors, it would appear that any AMPK agonist should produce the same phenotype. As cordycepin has already been shown to activate AMPK signaling (Wu et al. 2014, which has NOT been referenced in this manuscript) it is unclear why the authors have focused on this compound, as this finding is not novel. The generality of the AMPK-autophagy-Keap1-Nrf2 pathway should be demonstrated using other agonists of AMPK, such as AICAR, in the mouse model.

Minor comment

On line 175-176, the authors state that phosphorylation of Nrf2 on Serine 40 is "required" for Nrf2 translocation to the nucleus. This is not a well-established fact, and phosphorylation of Ser40 is rarely examined in the literature. The word "required" should be replaced with "implicated".

Reviewer #3 (Remarks to the Author):

The authors have significantly improved the quality of the paper and responded to the critiques raised by the reviewers.

Responses to reviewers:

We thank the reviewers for their thoughtful comments as well as the opportunity to respond. We have added new data to the paper as suggested and revised the paper to specifically address all the concerns raised by the editors and the reviewers. We have marked the changes in red and also modified the formats according to the checklist provided by the journal. We hope that the revised manuscript will now be acceptable for publication in Nature Communications.

Our responses to the reviewers' comments are enumerated below:

Reviewer #1:

Major points:

The authors have responded to all the issues raised by this reviewer. Additional experiments, particularly follow-up on overall survival of mice treated with cordycepin and showing significant increase in overall survival bolster considerably the overall impact. Moreover, addition of important 0 time point controls for proliferation and apoptosis effects of in IR or IR+ cordycepin groups raise confidence for the validity of the in vitro observations. Collectively, the results presented in this manuscript make a strong argument for radiation mitigation effects of cordycepin and have important clinical ramifications.

RE: We thank the reviewer for the comment.

Reviewer #2:

1. The relationship between radiation-induced DNA damage, SASP, AMP Kinase phosphorylation, and Nrf2 activity have not been adequately addressed in the mouse model. In Fig 4a-c, additional panels should be added to show the effect of Nrf2 inhibition (cordycepin plus ML385) and AMP kinase inhibition (cordycepin plus compound C) on DNA damage (γ

-H2AX) in response to radiation. Similarly, Fig 4e-f should also be expanded to include cordycepin plus ML385 and cordycepin plus compound C, in order to confirm the role of AMPK and Nrf2 in cellular senescence (p16 and p21 Western blots) and the SASP phenotype in the mouse model, and not merely in cell lines. This additional data would also clearly show the causality between AMPK, Nrf2, DNA damage and SASP in the mouse radiation model, as this is a critical factor which is currently lacking.

RE: We appreciate the reviewer for the constructive comment.

As suggested, we have added additional panels to show the effect of NRF2 inhibition (cordycepin plus ML385) and AMP kinase inhibition (cordycepin plus compound C) on DNA damage (γ -H2AX) in response to radiation (Fig. 4a-c).

And, we also added additional data to confirm the role of AMPK and NRF2 in cellular senescence and the SASP phenotype in the mouse model (supplementary figure 5a-f).

2. On line 71, the authors claim that they carried out a screen to identify cordycepin, yet no data or details of the screen are given within the manuscript. From the mechanism proposed by the authors, it would appear that any AMPK agonist should produce the same phenotype. As cordycepin has already been shown to activate AMPK signaling (Wu et al. 2014, which has NOT been referenced in this manuscript) it is unclear why the authors have focused on this compound, as this finding is not novel. The generality of the AMPK-autophagy-Keap1-Nrf2 pathway should be demonstrated using other agonists of AMPK, such as AICAR, in the mouse model.

RE: We thank the reviewer for the constructive comment.

1) Regarding the details of the screen, because there is an urgent need to develop effective mitigator for the radiation induced ulcer. In this study, we aim to identify new candidate to mitigate radiation ulcer.

First, because fibroblasts play vital roles in the development of ulcer, we established an in vitro

fibroblast cell senescence model induced by radiation to screen the candidate agents with a special attention on the chemical small molecules which can inhibit cell senescence;

Next, we used a library of small molecules established in our group to screen the potential radio-protective candidates for further investigations. Among the small molecules we tested, we identified cordycepin, a natural nucleoside analogue compound, as an effective mitigator to reduce the cell senescence and radiation-induced ulcer as shown in figure 2d-e, supplementary figure2a-e and figure 4d.

2) Regarding the phenotype of mitigating radiation ulcer by AMPK agonists, it has been demonstrated that despite different compounds activating the same target, the effect maybe varies because of the nature and timing of the activation (Derek T Hall et al, 2017).

As suggested by the reviewer, we applied two kinds of AMPK agonists (AICAR and metformin) to verify the generality of AMPK-autophagy-Keap1-NRF2 pathway in mitigating radiation ulcer. Both AICAR and metformin showed increased AMPK phosphorylation at Thr172 (p-AMPK) (Supplementary Fig11a), but their mitigating effects varies. The AMP-mimic AICAR, was able to prevent skin ulcer slightly, but was not as good as cordycepin (Supplementary Fig11b), while metformin failed to prevent skin ulcer.

These differences may be explained by the different mechanisms of AICAR and metformin to activate AMPK (Andrzejewski et al, 2014; Bridges et al, 2014; Grahame Hardie, 2016), further supporting that different activator of AMPK possess various biological activities. In this study, we first identified that cordycepin activate AMPK through interacting with the $\alpha 1$ and $\gamma 1$ subunit of AMPK near the autoinhibitory domain which directly relieve autoinhibition, which is different from either AICAR (activate AMPK by binding to the γ and β subunits) or metformin (directly inhibit complex I in the electron transport chain).

3) Regarding the effect of cordycepin to activate AMPK signaling, we thank the reviewer for the informative reference (Wu et al. 2014). In Wu 's work, cordycepin was identified to activate AMPK via interaction with the $\gamma 1$ subunit only in lipid regulation of HepG2 cancer cells, here, in our study,

we further identified cordycepin can activate AMPK through the interaction with both the $\alpha 1$ and $\gamma 1$ subunit of AMPK near the autoinhibitory domain which directly relieve autoinhibition and this new mechanism is demonstrated to play an important role in the prevention of cell senescence and mitigation of radiation ulcer. As suggested, we have cited and discussed Wu 's work in the revised manuscript (On line 379).

4) Regarding why we focused on cordycepin, because there is lack of safe and effective protective agents for the management of the radiation induced ulcer, and we have screened cordycepin as an effective mitigator to reduce the cell senescence and radiation-induced ulcer for the first time in our study. And because cordycepin is a natural compound and its safety has been verified by numerous investigations, representing a promising agent for future potential clinical application (Li Y et al.2016; Olatunji OJ et al.2016; Dou C et al.2016). This promotes us to further performed the investigation of cordycepin in radiation ulcer.

3. On line 175-176, the authors state that phosphorylation of Nrf2 on Serine 40 is “required” for Nrf2 translocation to the nucleus. This is not a well-established fact, and phosphorylation of Ser40 is rarely examined in the literature. The word “required” should be replaced with “implicated”.

RE: We appreciate the reviewer for the comment. As suggested, we have replaced the word “required” with “implicated” (on line 178).

Reviewer #3:

The authors have significantly improved the quality of the paper and responded to the critiques raised by the reviewers.

RE: We thank the expert for the review.

REVIEWERS' COMMENTS:

Reviewer #2 (Remarks to the Author):

Thank you for providing me a chance to review this paper again. I feel this study has reached a publication level although detail of the screen is still not sufficiently given within the manuscript. I also wrote a minor comment, which may be handled by the editor level.

Responses to reviewers:

We thank the reviewers for their thoughtful comments as well as the opportunity to respond. We have revised the paper to specifically address all the concerns raised by the editors and the reviewers. We have modified the formats according to the checklist provided by the journal. We hope that the revised manuscript will now be acceptable for publication in Nature Communications.

Our responses to the reviewers' comments are enumerated below:

Reviewer #2:

Thank you for providing me a chance to review this paper again. I feel this study has reached a publication level although detail of the screen is still not sufficiently given within the manuscript. I also wrote a minor comment, which may be handled by the editor level. Namely, from Figures 3 onwards, the authors should fully define what "control", "prevention" and "treatment" mean, and in the legend to Figure 4, they should say what ML385 and Compound C are inhibiting, as both of these points would make the manuscript easier for the reader to understand without having to refer back to the main text.

RE: We appreciate the reviewer for the constructive comment. As suggested by the reviewer, we have fully defined what "control", "prevention" and "treatment" mean in Figure 3, we also have explained what ML385 and Compound C are inhibiting in Figure 4.